# *Lkb1* inactivation drives lung cancer lineage switching governed by Polycomb Repressive Complex 2

Haikuo Zhang[1,2,\*], Christine Fillmore Brainson[3,4,5,\*,†], Shohei Koyama[1,2], Amanda J. Redig[1,2], Ting Chen[1,2], Shuai Li[1,2], Manav Gupta[3,4,5], Carolina Garcia-de-Alba[3,4,5], Margherita Paschini[3,4,5], Grit S. Herter-Sprie[1,2], Gang Lu[1,2], Xin Zhang[1,2], Bryan P. Marsh[3], Stephanie J. Tuminello[6], Chunxiao Xu[1,2], Zhao Chen[1,2], Xiaoen Wang[1,2], Esra A. Akbay[1,2], Mei Zheng[2], Sangeetha Palakurthi[7], Lynette M. Sholl[1,2], Anil K. Rustgi[8], David J. Kwiatkowski[1,2], J Alan Diehl[9], Adam J. Bass[1,2], Norman E. Sharpless[10], Glenn Dranoff[1,2], Peter S. Hammerman[1,2], Hongbin Ji[11,12], Nabeel Bardeesy[13], Dieter Saur[14,15], Hideo Watanabe[7], Carla F. Kim[3,4,5] & Kwok-Kin Wong[1,2,7,16]

Adenosquamous lung tumours, which are extremely poor prognosis, may result from cellular plasticity. Here, we demonstrate lineage switching of KRAS + lung adenocarcinomas (ADC) to squamous cell carcinoma (SCC) through deletion of *Lkb1* (Stk11) in autochthonous and transplant models. Chromatin analysis reveals loss of H3K27me3 and gain of H3K27ac and H3K4me3 at squamous lineage genes, including *Sox2, ΔNp63* and *Ngfr*. SCC lesions have higher levels of the H3K27 methyltransferase EZH2 than the ADC lesions, but there is a clear lack of the essential Polycomb Repressive Complex 2 (PRC2) subunit EED in the SCC lesions. The pattern of high EZH2, but low H3K27me3 mark, is also prevalent in human lung SCC and SCC regions within ADSCC tumours. Using FACS-isolated populations, we demonstrate that bronchioalveolar stem cells and club cells are the likely cells-of-origin for SCC transitioned tumours. These findings shed light on the epigenetics and cellular origins of lineage-specific lung tumours.

[1] Department of Medical Oncology, Dana-Farber Cancer Institute, Boston, Massachusetts 02215, USA. [2] Department of Medicine, Brigham and Women's Hospital, Harvard Medical School, Boston, Massachusetts 02115, USA. [3] Stem Cell Program and Division of Hematology/Oncology, Boston Children's Hospital Boston, Boston, Massachusetts 02115, USA. [4] Department of Genetics, Harvard Medical School, Boston, Massachusetts 02115, USA. [5] Harvard Stem Cell Institute, Cambridge, Massachusetts 02138, USA. [6] Department of Medicine, Division of Pulmonary, Critical Care and Sleep Medicine; Tisch Cancer Institute, Icahn School of Medicine at Mount Sinai, New York, New York 10029, USA. [7] Belfer Institute for Applied Cancer Science, Dana-Farber Cancer Institute, Boston, Massachusetts 02215, USA. [8] University of Pennsylvania Perelman School of Medicine, Division of Gastroenterology, Department of Medicine and Genetics, Abramson Cancer Center, Philadelphia, Pennsylvania 19104, USA. [9] Department of Biochemical and Molecular Biology, Medical University of South Carolina, Charleston, South Carolina 29425, USA. [10] University of North Carolina Lineberger Comprehensive Cancer Center, UNC School of Medicine, Chapel Hill, North Carolina 27599, USA. [11] Key Laboratory of Systems Biology, CAS Center for Excellence in Molecular Cell Science, Innovation Center for Cell Signaling Network, Institute of Biochemistry and Cell Biology, Shanghai Institutes for Biological Sciences, Chinese Academy of Sciences, Shanghai 200031, China. [12] School of Life Science and Technology, Shanghai Tech University, Shanghai 200120, China. [13] Cancer Center, Massachusetts General Hospital, Boston, Massachusetts 02144, USA. [14] Department of Internal Medicine II, Klinikum rechts der Isar, Technische Universität München, München 81675, Germany. [15] German Cancer Research Center (DKFZ) and German Cancer Consortium (DKTK), Im Neuenheimer Feld 280, 69120 Heidelberg, Germany. [16] Laura and Isaac Perlmutter Cancer Center, New York University Langone Medical Center, New York, New York 10016, USA. \* These authors contributed equally to this work. † Present address: Department of Toxicology and Cancer Biology, University of Kentucky, Lexington, Kentucky 04536, USA. Correspondence and requests for materials should be addressed to C.F.K. (email: Carla.kim@childrens.harvard.edu) or to K.-K.W. (email: kwok-kin.wong@nyumc.org).

L ung adenocarcinoma (ADC) and squamous cell carcinoma (SCC) are regarded as segregated entities of non-small-cell lung cancer. These two tumour types largely have unique genetic profiles, with KRAS and EGFR activation common in ADC, while *PTEN* inactivation, PI3KCA activation and NOTCH2 activation are common in SCC tumours[1,2]. However, there are cases of mixed histology tumours, where at least 10% of the biopsied tumour cells have one histology (either ADC or SCC) and the remainder are the other histology[3]. Adenosquamous (ADSCC) disease accounts for 2–3% of all diagnoses, though this diagnosis can only be made with wedge biopsy or tumour resection, which could lead to underestimation of its prevalence in patient populations[3,4]. ADSCC is a particularly poor prognosis tumour type[5,6], and genetic studies have demonstrated that both histological components share mutations, suggesting a monoclonal tumour origin[7]. Notably, in many cases of disease progression after targeted therapy for EGFR mutation, when it is clinically justifiable to take a second biopsy, conversion of ADC to SCC has been observed[8]. Given these data, a better understanding of lung cancer lineage relationships could shed light on both the origins of lung cancer and how to overcome therapeutic resistance.

SCCs have long been proposed to arise from tracheal basal cells and ADCs have been proposed to arise from alveolar type II (AT2) cells or club (Clara) cells, due to markers of these cell types being present in the malignant lesions[4,9]. However, given the shared genetics of ADC and SCC lesions in ADSCC tumours, it must be possible for certain lung cells to drive both histologies. Basal cells, which express nerve growth factor receptor (NGFR), p63 and cytokeratin 5 (KRT5), serve as stem cells for the trachea, main bronchi and upper airways. Basal cells can replace the pseudostratified epithelium including secretory club cells, mucus-producing goblet cells and ciliated cells[10–12]. In more distal airways, club cells are a self-renewing population that maintain the ciliated cells[13]; subsets of club cells can give rise to ciliated and club cell lineages after injury[14,15]. In the alveolar space where gas exchange is carried out by alveolar type I cells, the surfactant-expressing AT2 cells act as stem cells[16,17]. Cells expressing club cell secretory protein (CCSP), including bronchioalveolar stem cells (BASCs), can give rise to AT2 cells[18–22]. There is also extensive plasticity in the lung and tracheal epithelium, as club cells can give rise to basal cells[23], and may give rise to KRT5+/p63+ cells or alveolar cells under certain injury conditions[24,25]. Cellular lineage switching, either in the normal situation or in cancer, could be modulated by epigenetic mechanisms, including histone modification governed in part by the Polycomb Repressive Complex 2 (PRC2).

Genetically engineered mouse models are unparalleled in their capacity to allow the study lung tumour origins and evolution. Using a *LSL:Kras*[G12D/+]; *Lkb1*[flox/flox] (LSL = Lox-stop-Lox) mouse model of lung cancer, we demonstrated previously that *Lkb1* inactivation dramatically accelerated KRAS-driven lung cancer progression and changed the tumour spectrum from purely ADC to ADC and SCC[26]. While KRAS is a common oncogene in lung ADC, *LKB1/STK11*, encoding a serine-threonine kinase implicated in energy sensing and cell polarity, is notable as being among the most commonly mutated tumour suppressors in ADC[2]. Additionally, mutations in *LKB1* predominantly co-occur with *KRAS* activating mutations[27,28]. Subsequent studies with the *LSL:Kras*[G12D/+]; *Lkb1*[flox/flox] mouse model demonstrated that the SCC tumours arise later during tumour progression than ADC and that SCCs are characterized by decreased lysyl oxidases and increased reactive oxygen species[29–31]. However, because of the simultaneous activation of KRAS and inactivation of *Lkb1*, it remained unclear if a unique pool of SCC competent cells were transformed only

when *Lkb1* was deleted, or if existing KRAS-induced ADC could convert to a squamous fate in response to *Lkb1* deletion. Furthermore, due to the intranasal inhalation method to introduce Cre to drive the genetics, the cell-of-origin of this tumour type was unknown.

Here, we describe a stepwise mouse model of lung tumorigenesis that strongly supports the theory that established ADC cells can transition to SCC fate upon additional genetic perturbations, such as *Lkb1* deletion. Using this model, we found that de-repression of squamous genes through loss of Polycomb-mediated gene repression accompanies the squamous transition. We also show that club cells and BASCs are the most fit populations to give rise to adenosquamous tumours. Together these data add to our understanding of the underlying epigenetic programmes and cellular origins of lineage-specific lung tumours.

## Results

**Lkb1 deletion drives SCC transition of established KRAS ADCs.** Previously, we showed that *Stk11* (*Lkb1*) deletion concomitant with induction of oncogenic KRAS drove acquisition of aggressive tumour characteristics, including SCC transition, not observed in KRAS tumours when *Lkb1* is intact[26]. These data were confounded by the fact that *Lkb1* mutations are relatively infrequent in pure SCC tumours (2%, see ref. 1). However, the model of KRAS and *Lkb1* is actually a mixed histology model, containing ADC, SCC and mixed ADSCC tumours. Thus, we hypothesized that *LKB1* mutations may be more frequent in patient lung ADSCC samples. Data from a published study[32] and from a cohort of ADSCC tumours at DFCI showed that of 23 ADSCC tumour cases, 6 harboured *LKB1* inactivation. These data suggest that *LKB1* mutations may be more frequent (26.1%), or at least as frequent, in ADSCC tumours as in ADC tumours (15.6%, 111 of 602 ADC tumours, $P = 0.24$). Together with data demonstrating KRAS is often activated in ADSCC tumours[33], we propose that ADSCC in KRAS/*Lkb1* mice is a clinically relevant recapitulation of the genetics found in human lung adenosquamous patients.

To dissect the contributions of KRAS and *Lkb1* mutations during lung tumorigenesis, we used the dual recombinase model to activate oncogenic KRAS and vary the time-point of *Lkb1* deletion. Our model includes that alleles *FSF:Kras*[G12D/+]; *FSF:R26:CreERT2*; *Lkb1*[flox/flox] (FSF = Fret-Stop-Fret) and combines flippase-FlpO-Fret and tamoxifen (Tam) inducible Cre-loxP recombination to make KRAS[G12D] activation and *Lkb1* inactivation separable (Fig. 1a). Mice were randomized into four arms: (1) KRAS activation with *Lkb1* deletion at the same time (concomitant), (2) KRAS activation followed by *Lkb1* deletion after 2 weeks, (3) KRAS activation followed by *Lkb1* deletion after 10 weeks and (4) KRAS activation alone (Fig. 1a). Compared to mice with KRAS activation alone, overall survival of the cohorts was shortest when *Lkb1* was deleted concomitant with KRAS activation, and was significantly shortened when *Lkb1* was deleted either 2 weeks or 10 weeks after KRAS activation ($P = 0.0081$ KRAS alone versus 10 weeks, Fig. 1b). We confirmed that tamoxifen administration was causing bi-allelic deletion of *Lkb1* by performing western blot on whole tumour extracts (Supplementary Fig. 1a). In addition to accelerated tumour growth and reduced survival, lymph node and distant tumour metastases were present in all three cohorts in which *Lkb1* was deleted (Supplementary Fig. 1b). We next sought to determine if deletion of *Lkb1* long after KRAS activation could drive the increased histology spectrum observed previously[26]. In the two cohorts where *Lkb1* was deleted after induction of KRAS, we observed acquisition of squamous characteristics to the same degree as observed with the concomitant model, with

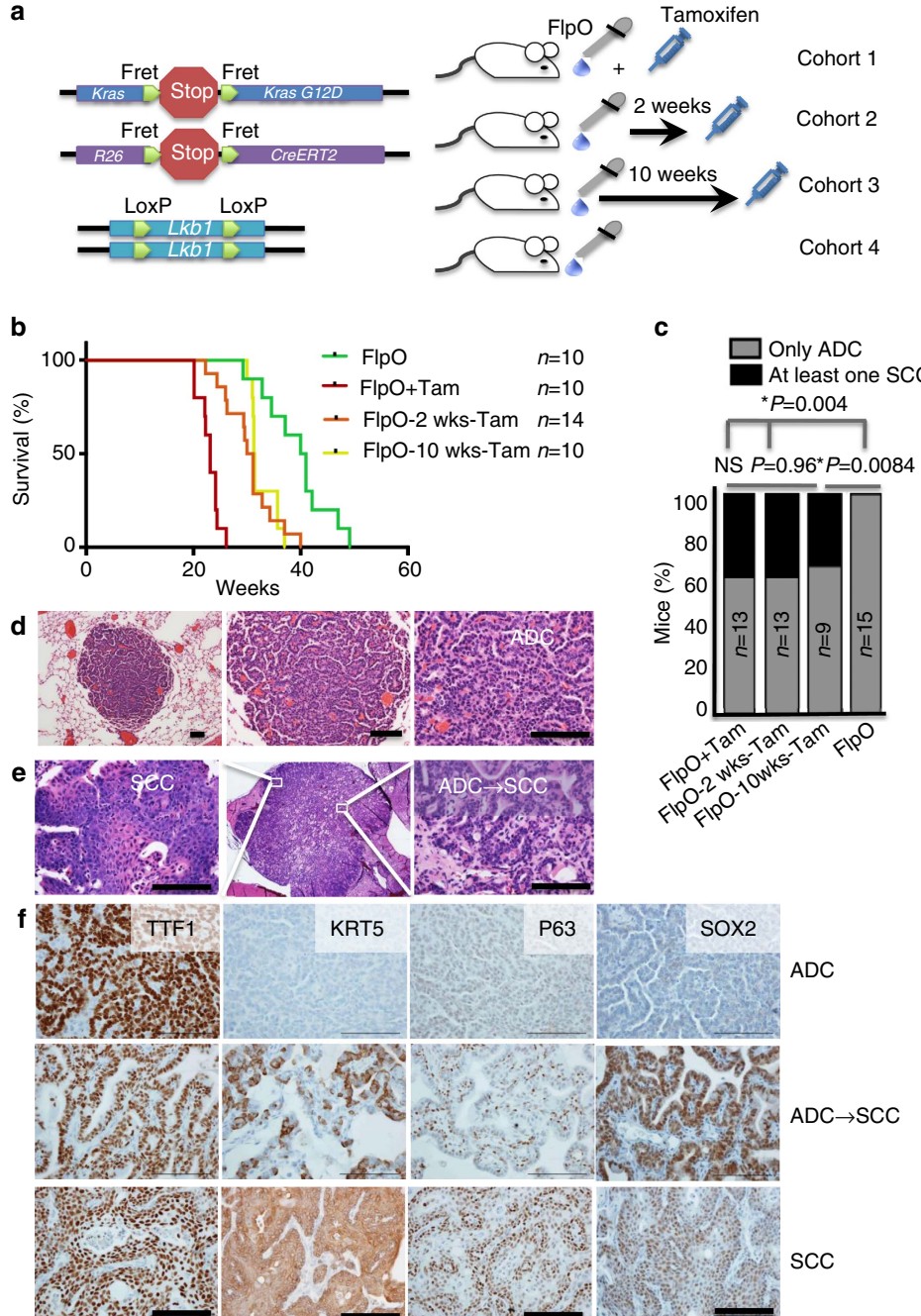

**Figure 1 | *Lkb1* deletion drives SCC transition and tumour progression in established KRAS tumours.** (**a**) Schematic of the four cohorts of mice used in this study: (1) KRAS activation with *Lkb1* deletion at the same time (FlpO + Tam), (2) KRAS activation followed by *Lkb1* deletion after 2 weeks (FlpO-2weeks-Tam), (3) KRAS activation followed by *Lkb1* deletion after 10 weeks (FlpO-10 weeks-Tam) and (4) KRAS activation alone (FlpO). (**b**) Kaplan–Meier survival of the four cohorts, *n* indicated in the figure, $P < 0.0001$ between FlpO + Tam and 2-week Tam, $P = 0.0081$ between 10-week Tam and FlpO. *P* values in **b** represent log-rank test. (**c**) Percentage of mice with at least one purely squamous lesion as determined by histology at end point in the four cohorts, *n* and *P*-values indicated in the figure, *P*-values represent $\chi^2$-test. (**d**) Haematoxylin and eosin staining of tumours at 10 weeks post FlpO show only adenocarcinoma histology. Scale bar, 50 μm. (**e**) Haematoxylin and eosin staining of tumours at 20 weeks post tamoxifen shows tumours that are undergoing transition from adenocarcinoma to squamous cell carcinoma histology, scale bar, 50 μm. (**f**) Validation of tumour subtyping by immunohistochemistry for TTF1 (NKX2.1), KRT5, p63, and SOX2. Scale bar, 50 μm. See also Supplementary Fig. 1a–c.

approximately 40% of mice harbouring at least one tumour with purely squamous characteristics at the end of the study ($P = 0.96$ concomitant versus 10 week; Fig. 1c). Consistent with previous observations[34], tumours that develop when KRAS is activated by FlpO were uniformly low-grade lung tumours that always present as glandular adenomas and ADCs (Fig. 1d).

In *Lkb1*-deleted tumours, purely squamous and transitioning tumours with areas of both ADC and SCC histology, as determined by immunohistochemistry (IHC) for the squamous markers KRT5, p63 and SOX2, were present (Fig. 1e,f). By contrast, the distal lung marker TTF1 (NKX2.1) was highest in the ADC lesions, and decreased during squamous transition.

Consistent with the microenvironment also being affected[35,36], we observed that the tumour-associated myeloid cells changed during transition from ADC to SCC. As previously observed[37], ADC lesions stained positively for the macrophage marker F4/80, while the SCC lesions stained positively for MPO, a marker of tumour-associated neutrophils (Supplementary Fig. 1c).

Our first set of experiments demonstrated that *Lkb1* deletion in cells that had KRAS activation long before was sufficient to produce tumours with squamous characteristics. It was tempting to speculate that this was evidence of a true transition from a KRAS+ ADC cellular state to a KRAS+/*Lkb1-null* squamous state. However, because the two genetic manipulations were performed on cells that remained within the same mouse during the entirety of the process, the possibility remained that *Lkb1* deletion was allowing growth of a latent population of squamous-predisposed KRAS+ cells. To address this issue, we isolated KRAS+ lung ADC cells prior to tamoxifen treatment and transplanted them into immunocompromised mouse recipients (Fig. 2a). Cohorts of mice received either tamoxifen or placebo were aged until signs of tumour distress, and were then assessed for tumour histology. The tamoxifen naïve cohort uniformly presented with ADC, which could be serially transplanted and never transitioned to SCC (Fig. 2b). In contrast, upon tamoxifen administration 2 weeks after transplant, the tumours in mice treated with tamoxifen took on squamous characteristics, with 40% of mice having at least one lesion that was completely SCC at the time of killing ($P = 0.0225$ Tam versus no Tam, Fig. 2c). In our previous studies, we observed that KRAS tumour propagating cells (TPCs) were present in both Sca1+ and Sca1- compartments, and a purely squamous model generated by bialleleic inactivation of *Lkb1* and *Pten* (*Lkb1/Pten*) harboured Sca1+ and NGFR+ TPCs[37]. To explore these populations in the transplanted tumours, we collected tumours from each mouse, dissociated and analysed EpCAM+/CD31−/CD45− cells for expression of NGFR and Sca1. We observed that KRAS transplant tumours had very few Sca1+/NGFR+ cells and were indistinguishable from adeno-Cre induced KRAS tumours ($P = 0.368$; Fig. 2d and Supplementary Fig. 2a,b). By contrast, tamoxifen-treated mice harboured tumours with high levels of Sca1 and NGFR, and the population of Sca1+/NGFR+ cells was the same proportion of tumour as found in the pure *Lkb1/Pten* SCC tumours ($P = 0.004$ Tam versus no Tam, $P = 0.720$ Tam versus *Lkb1/Pten*). We also confirmed cell surface NGFR staining in the tamoxifen-treated mouse tumours by IHC (Supplementary Fig. 2c). Lastly, we isolated cells from squamous transitioned lesions for transplant, and found that both Sca1^High NGFR^High and Sca1^High NGFR^Low tumour cells could successfully transplant squamous disease (Fig. 2e). Together these data demonstrate that purely ADC lesions derived from transplanted KRAS+ cells can transition to SCC, and that tumours can acquire the SCC TPC expression profile of Sca1+/NGFR+ upon *Lkb1* deletion. Once the squamous transition has happened, the phenotype perpetuates.

**Loss of PRC2 activity accompanies SCC transition**. We hypothesized that the switch from ADC to SCC could be controlled in part through epigenetic mechanisms. The decreased, yet not absent, expression of TTF1 in the SCC lesions could be indicative of epigenetic memory of prior TTF1 positivity before the squamous differentiation process occurred, as was observed in other studies[30]. To examine the possibility that an epigenetic mechanism such as PRC gene repression was involved in the ADC-SCC transition, we performed western blots for the common methylation marks catalysed by PRC, H3K27me1,

H3K27me2 and H3K27me3, on whole tumour extracts from KRAS-induced ADCs (no tamoxifen), KRAS/*Lkb1* ADCs and KRAS/*Lkb1* SCCs (both tamoxifen at 10 weeks post FlpO). When compared to KRAS ADC or KRAS/*Lkb1* ADC, there was a marked increase in histone H3 lysine 27 mono-methylation (H3K27me1) accompanied by a decrease in H3K27me3 in the KRAS/*Lkb1* SCC lesions (Fig. 3a). We next examined expression of the components of the PRC2, the protein complex that catalyses the H3K27me3 mark, by western blotting. Interestingly, although SCC lesions have higher levels of the methyltransferase EZH2 than the ADC lesions, there is a clear lack of the essential PRC2 subunit EED in the SCC lesions (Fig. 3b). In a published data set from KRAS ADC, KRAS/*Lkb1* ADC and KRAS/*Lkb1* SCC[31], transcriptional levels of EED were not significantly different, suggesting regulation of protein stability. To explore this inverse correlation between EZH2 and H3K27me3 expression, we performed immunostaining for the two markers on serial tumour sections from KRAS/*Lkb1* adenosquamous tumours, *Pten/Lkb1/p53* adenosquamous tumours and *Pten/Lkb1* squamous tumours (Fig. 3c). Consistently, we observed that ADC regions had sparse EZH2 staining and robust H3K27me3 staining. In contrast, squamous lesions on the same slides showed higher levels of EZH2 staining, and lower, but not absent, H3K27me3 staining (Fig. 3d and Supplementary Fig. 3a). We performed similar staining on human tumours, including six confirmed cases of adenosquamous cancer (Fig. 3e). Again, the pattern of higher EZH2 and slightly lower, but not completely absent, H3K27me3 was present in squamous lesions as compared to ADC lesions, which were H3K27me3 high and had lower levels of EZH2 (Fig. 3f and Supplementary Fig. 3b). While analysing these sections, we noted that normal airway and alveolar epithelium shared the ADC staining pattern of high H3K27me3 and low EZH2, while the oesophagus, which is the closest stratified squamous organ to the lung, had increased EZH2 staining and lower H3K27me3 (Supplementary Fig. 3b). This observation implies that normal lung epithelium may be maintained in a glandular state by PRC2 gene repression, and that loss of H3K27me3 gene repression may de-repress a programme predisposed to the squamous cell sate.

Despite the obvious lack of EZH2's catalytic mark, higher levels of EZH2 in the squamous tumours suggested that EZH2 may be required for squamous tumour growth and could represent a therapeutic target. Therefore, we bred the *Ezh2* floxed alleles[38] to the Cre-based *LSL:KrasG12D*; *Lkb1*^flox/flox model. We generated three genotypes with the *Kras/Lkb1* alleles in the setting of *Ezh2*^+/+, *Ezh2*^flox/+ or *Ezh2*^flox/flox alleles (Fig. 4a). Cohorts of these genotypes were analysed for tumour histology 11 weeks post Adeno-Cre virus administration. While the *Ezh2*^+/+ and *Ezh2*^flox/+ cohorts both had ~40% of mice with at least one purely squamous lesion at this time-point, all of the mice from the *Ezh2*^flox/flox cohorts had squamous lesions (Fig. 4b,c). We confirmed that both H3K27me3 and EZH2 were decreased in these *Ezh2*^flox/flox tumours by IHC (Supplementary Fig. 4a,b). This result suggests that EZH2 is dispensable for the squamous state, and that loss of the PRC2 complex activity could potentiate the switch to SCC. This result is unique from those observed when the PRC2 complex was depleted in KRAS or KRAS/*p53* tumours—without *Lkb1* deletion the squamous phenotype does not appear[39].

The genetic results described above suggest that loss of EZH2 function can perpetuate the squamous fate over several weeks of tumour development. We next wanted to test if inhibiting the catalytic function of EZH2 with a small molecule could drive the acquisition of squamous markers in short-term *in vitro* cultures. We first treated the human ADC cell line A549, which has the genotype of KRAS^G12S, LKB1^Q37* with the inhibitor

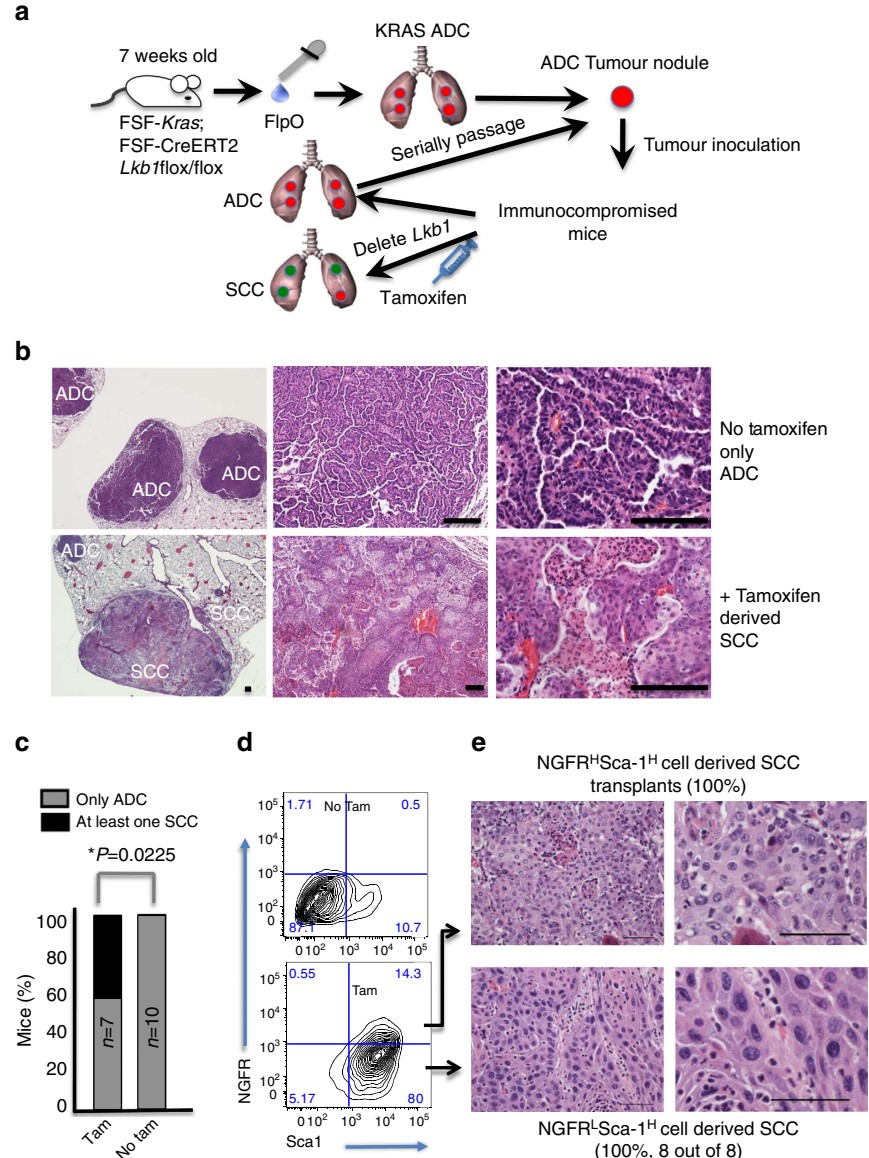

**Figure 2 | *Lkb1* deletion after transplant of KRAS adenocarcinoma cells drives SCC transition.** (**a**) Schematic of adenocarcinoma induction, transplantation and deletion of *Lkb1* in recipient mice. (**b**) Haematoxylin and eosin staining of transplanted KRAS tumours show only adenocarcinoma histology, while ADSCC and SCC were visible in tamoxifen-treated mice, scale bar, 50 μm. (**c**) Percentage of mice with at least one purely squamous lesion as determined by histology at end point in the four cohorts, *n* and *P* values indicated in the figure, *P* values represent $\chi^2$-test. (**d**) Representative flow cytometric plot of dissociated tumours from untreated (no tamoxifen) transplanted KRAS tumours (top) and tamoxifen-treated mice (bottom) gated on DAPI$^-$/CD31$^-$/CD45$^-$/EpCAM$^+$ cells and showing NGFR and Sca1 staining. (**e**) Haematoxylin and eosin staining tumours derived from transplanted Sca1$^{High}$NGFR$^{High}$ or Sca1$^{High}$NGFR$^{Low}$ cells from an SCC KRAS/*Lkb1* lesion demonstrates that the switch to the squamous phenotype is stable, scale bar, 50 μm. See also Supplementary Fig. 2a–c.

GSK126 (ref. 40). As measured by flow cytometry, the NGFR + cells within this cell line increased from 4.9 to 20.4% after 6 days in 10 μM GSK126 (*P* < 0.0001, Fig. 4d). RNA from the treated A549 cells likewise showed a fourfold increase in NGFR transcript level in response to GSK126 relative to vehicle-treated cultures, consistent with a de-repression of EZH2-mediated silencing of the locus (*P* = 0.0004; Fig. 4e). Although the *LSL:KrasG12D* cells that are WT for p53 are not able to grow as two-dimensional adherent cultures, they can be grown as three-dimensional (3D) cultures[37]. We confirmed that growing 3D tumour organoids derived from FlpO induced KRAS lesions in 100 nM 4-hydroxy tamoxifen was able to drive deletion of the *Lkb1* allele, making this a tractable system in which to observe the early changes associated with *Lkb1* loss (Supplementary Fig. 4c). Firstly, we observed an increase in organoid re-seeding ability with tamoxifen treatment, consistent with the acquisition of more aggressive tumour characteristics upon *Lkb1* deletion *in vivo* (*P* = 0.04; Fig. 4f and Supplementary Fig. 4d). Whereas KRAS tumour organoids were significantly decreased when passaged in continued 5 μM GSK126, organoids grown in the presence of tamoxifen (and therefore *Lkb1* null) were equally passaged in continued presence of the EZH2 inhibitor (*P* = 0.0113 for *Kras*, *P* = 0.132 for KRAS/*Lkb1*; Supplementary Fig. 4d). Lastly, we observed that treating cultures with tamoxifen for 9–12 days (primary organoids) and re-seeding the cells in drug for an additional 9–12 days (secondary organoids) led a marked

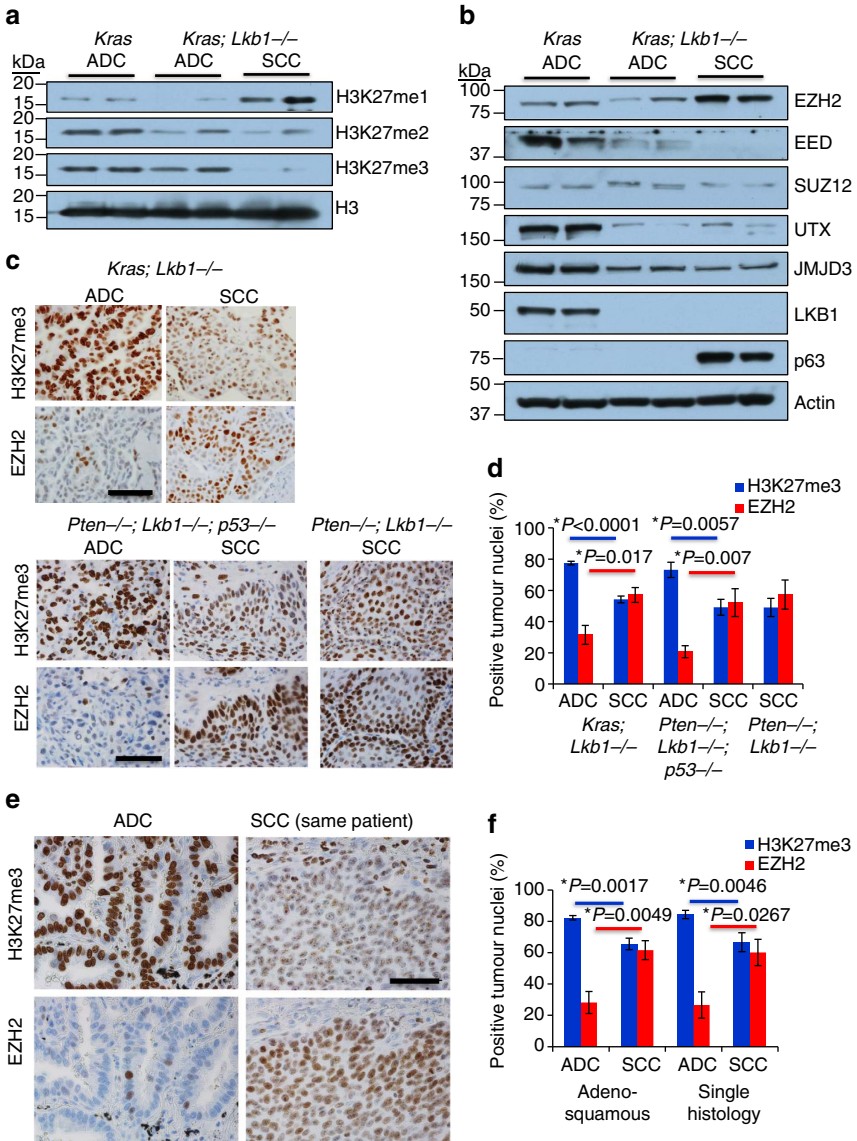

**Figure 3 | Loss of histone H3 lysine 27 trimethylation accompanies SCC transition.** (**a**) Western blotting analysis performed on whole-cell extracts from tumours of the indicated genotypes and histologies. Histone H3 lysine 27 tri-methylation (H3K27me3) is markedly lower in SCC lesions; total histone H3 is the loading control. (**b**) Western blotting analysis performed on whole-cell extracts from tumours of the indicated genotypes and histologies. p63 and LKB1 confirm the histologies and genotypes of the lysates, and while EZH2 in more highly expressed in SCC tumours, the essential PRC2 subunit EED is absent in the SCC lesions. β-Actin is the loading control. (**c**) Immunohistochemistry for H3K27me3 on a panel of mouse lung tumours of the indicated histologies from KRAS/*Lkb1*, *Pten/Lkb1/p53*, and *Pten/Lkb1* mice, scale bar, 50 μm. (**d**) Quantification of nuclear staining by dot analysis with Nikon software; data are mean ± s.e.m. measured on serially stained sections, $n = 6$–10. (**e**) Immunohistochemisty for H3K27me3 on a panel of human mixed adenocarcinomas, pure ADC and pure SCCs. Scale Bar, 50 μm. (**f**) Quantification of nuclear staining, for H3K27me3 $n = 6$ ADSCC, 14 ADC, 9 SCC, for EZH2 $n = 6$ ADSCC, 7 ADC, 5 SCC, data are mean ± s.e.m. measured on serially stained sections. *P* values represent 2 tailed t-test. See also Supplementary Fig. 3a–c.

decrease in *Lkb1* expression and an increase in *Sox2*, *Ngfr* and *Sca1* expression (Fig. 4g and Supplementary Fig. 4e). Interestingly, GSK126 treatment potentiated the increase in both *Sca1* and *Sox2*, consistent with loss of H3K27me3 transcriptional repression allowing the SCC transition process.

**Chromatin landscapes reveal de-repression of squamous genes.** To characterize the chromatin landscapes of KRAS/*Lkb1* tumours, we performed chromatin immunoprecipitation (ChIP) on microdissected tumours confirmed to be either ADC or SCC by histology and qPCR (Supplementary Fig. 5a). The two activating marks, histone H3 lysine 4 tri-methylation (H3K4me3)

and histone H3 lysine 27 acetylation (H3K27ac), and the PRC2-derived silencing mark, histone H3 lysine 27 tri-methylation (H3K27me3) were immunoprecipitated, followed by sequencing the chromatin bound DNA (ChIP-seq). ADC and SCC tumours could be clearly distinguished by differential H3K4me3 enrichment (Fig. 5a). Of the most significantly differential enrichment on H3K4me3 and/or H3K27ac marks between SCC and ADC tumours (Supplementary Fig. 5c and Supplementary Tables 1–4), we found much higher load of H3K27ac and H3K4me3 marks in the SCC tumour compared to the ADC tumour on the squamous genes *Sox2*, *ΔNp63*, *Ngfr* and *Krt5/6* (Fig. 5b and Supplementary Fig. 5a). Furthermore, we observed a significantly lower level of H3K27me3 mark in *Sox2*,

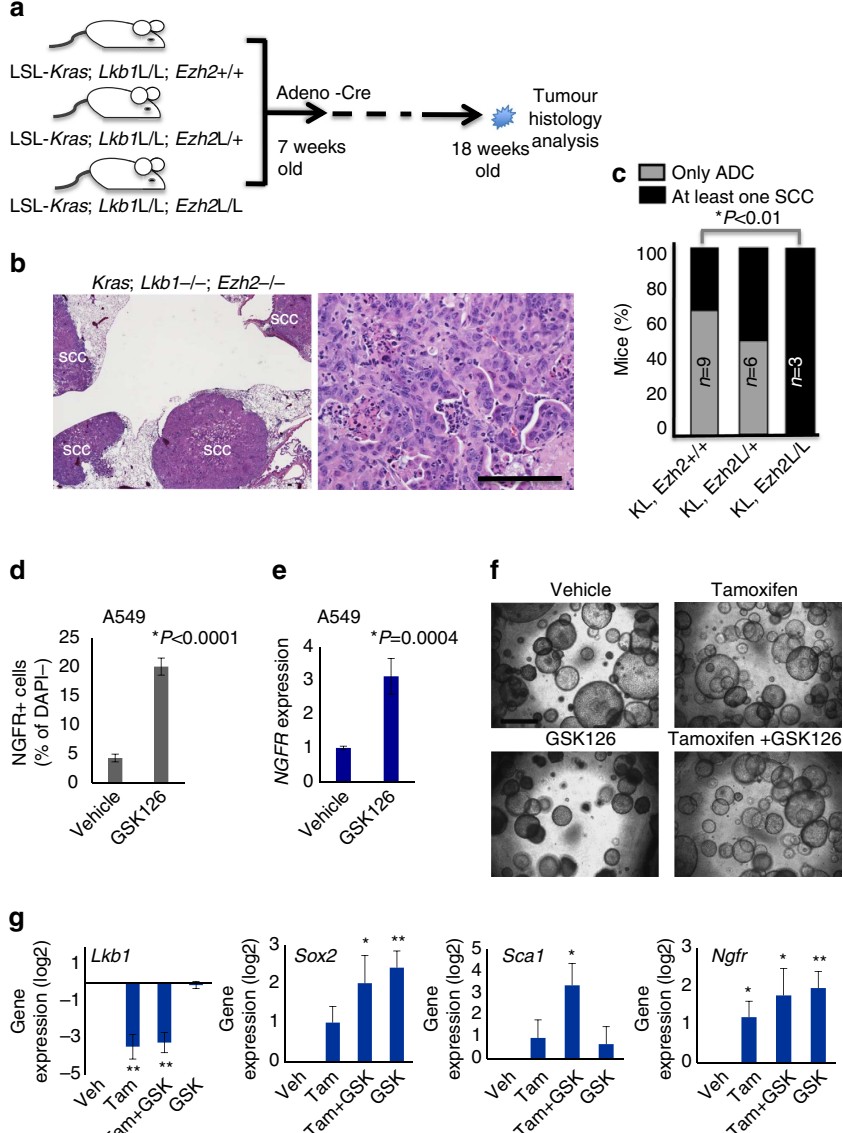

**Figure 4 | Deletion or inhibition of EZH2 potentiates SCC transition.** (**a**) Schematic of deletion of *Ezh2* in addition to KRAS activation and *Lkb1* deletion. (**b**) Haematoxylin and eosin staining of tumours at 11 weeks post Adeno-Cre show predominant SCC histology in the *Ezh2*fl/fl; *Kras*G12D; *Lkb1*fl/fl mice, scale bar, 50 μm. (**c**) Percentage of mice with at least one purely squamous lesion as determined by histology at end point in the four cohorts, *n* and *P* values indicated in the figure, *P* values represent $\chi^2$-test. (**d**) Flow for NGFR in the KRAS+/*LKB1-null* human line A549 treated 10 μM of EZH2 inhibitor GSK126 for 6 days, data are mean ± s.e.m., *n* = 4, *P* < 0.0001. (**e**) RT-qPCR for *NGFR* expression in the A549-treated lines, data are mean ± s.e.m., *n* = 4, *P* = 0.0004. (**f**) Representative images of indicated secondary tumour organoids plated at 20,000 cells per transwell, scale bar, 200 μm. (**g**) RT-qPCR for *Sox2*, *Sca1* and *Lkb1* in tumour organoid 3D cultures treated with 100 nM tamoxifen, 5 μM of EZH2 inhibitor GSK126, or both for 9–12 days, mean ± s.e.m. on $\log_2$ scale is graphed, **indicates *P* < 0.01, *indicates *P* < 0.05, *n* = 4. *P* values for **d**, **e** and **g** represent two-tailed *t*-test. See also Supplementary Fig. 4a–e.

*Ngfr* and *Krt5/6* loci compared to those in ADC tumours, consistent with de-repression of these squamous loci in SCC tumours (Fig. 5b). The differentially enriched regions for active histone marks also included the loci for the neutrophil chemoatractants *Cxcl3/5*, and the interferon-response induced *Ifitm1/2/3*, which were higher in the SCC tumours (Supplementary Fig. 5a). For ADC, the known expressed genes *Scgb1a1*, *Foxa2* and *SftpB* had activating marks which were lost is SCC (Supplementary Fig. 5b). We also used the ROSE algorithm[41] to call super-enhancers, many of which were shared between ADC and SCC, indicating that the tumours share some epigenetic memory (Supplementary Tables 5 and 6). For ADC, a unique super-enhancer was called at *Scgl1a1*, while *Ifitm3* was a unique super-enhancer for SCC (Supplementary

Tables 5 and 6). We next ran GREAT[42] on the significantly enriched H3K4me3 loci in SCC tumours and found that the genes adjacent to these loci are enriched for H3K27me3, methylated CpG islands and PRC2 components in human embryonic stem cells and murine embryonic fibroblasts, suggesting that the activated loci in SCC tumours are normally repressed by PRC2 in other cell types (Fig. 5c). A closer examination of the H3K27ac-enriched regions in each tumour type indicated that unlike the H3K27ac-marked loci that were common to all samples, which were devoid of H3K27me3, the uniquely SCC- or ADC-enriched loci were very often bivalently marked with H3K27me3 and H3K37ac in ADC, and became monovalently marked with H3K27ac in SCC due to a loss of PRC2-mediated gene repression (*Sox2* is an example) (Fig. 5d).

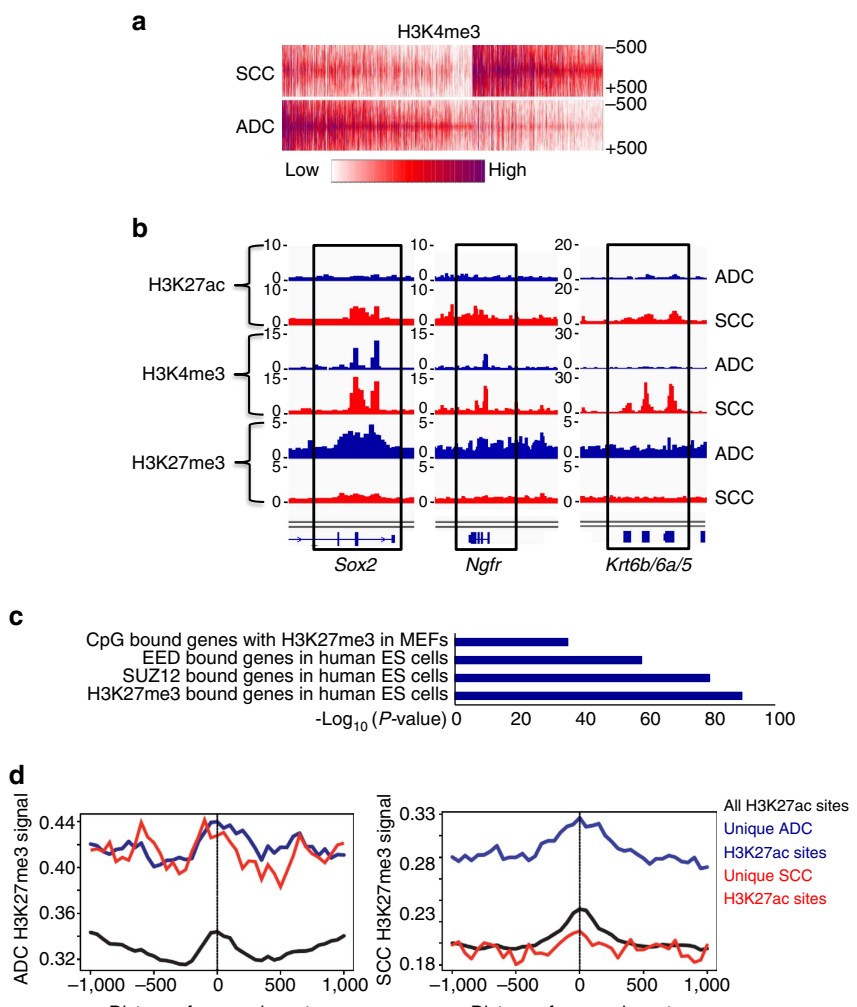

**Figure 5 | Chromatin landscapes of ADC and SCCs reveal de-repression of squamous genes.** (**a**) Heatmap depicting global analysis of H3K4me3 marked chromatin in ADC and SCC from KRAS/Lkb1 mice. (**b**) Genome browser snapshots of the squamous loci Sox2, Ngfr and Krt6b/6a/5 with the indicated ChIP-sequencing peaks for H3K27ac, H3K27me3 and H3K4me3 in the ADC (blue) and SCC (red) KRAS/Lkb1 tumour samples. (**c**) GREAT analysis of genes adjacent to H3K4me3-enriched sites specific to SCC showed that this gene set is enriched for genes (from MSigDB) that are normally repressed by Polycomb in other cell types. Graphed as -log10 (P value represents hypergeometric probability test.) Level of H3K27me3 marks at all loci enriched for H3K27ac (black), loci with H3K27ac unique to ADC (blue) or loci with H3K27ac unique to SCC (red) in either ADC (left panel) or SCC (right panel) from KRAS/Lkb1 mice. See also Supplementary Fig. 5a–c.

**BASCs and club cells are cells-of-origin for SCC transition.** We reasoned that the 'choice' of cells to transition to the squamous fate upon Lkb1 deletion may be predetermined by the tumour cell-of-origin. To test this theory, we used a fluorescence activated cell sorting (FACS) approach to enrich for populations of lung cells that are likely tumour cells-of-origin and which can be easily enriched using cell surface markers, including tracheal basal cells (enriched using positive selection for NGFR), non-basal cells of the trachea (the NGFR- fraction from the trachea that contains club, ciliated and goblet cells[23,43]), distal lung BASCs (enriched in the EpCAM+ Sca1+ fraction) and distal lung alveolar type II (AT2) cells (enriched in the EpCAM+ Sca1- fraction)[12,21] (Fig. 6a and Supplementary Fig. 6a). To directly test the fitness of each population to propagate after activation of oncogenic KRAS we incubated each FACS-isolated population with no virus, adeno-GFP virus or adeno-FlpO virus before plating for organoid cultures[12,20] (Fig. 6b and Supplementary Fig. 6b). Strikingly, while the non-basal cells, BASCs and AT2 populations were all able to give rise to organoids after KRAS activation (Fig. 6c–d and Supplementary

Fig. 6c), the basal cell population was consistently unable to produce any organoid after adeno-FlpO virus (Fig. 6c,d, n = 4). Visual inspection of the basal cell cultures showed single non-dividing cells several days after plating. The non-basal cells, despite having a dramatically decreased organoid forming efficiency compared to basal cells, were consistently able to grow robust KRAS+ organoids (Fig. 6c–e). Both non-basal and BASC-derived organoids were predominantly bronchiolar in phenotype, while AT2-derived organoids were alveolar (Supplementary Fig. 7a).

Having formed KRAS+ organoids with our in vitro culture system, we next sought to delete Lkb1 in these cultures and assess if organoids could take on squamous characteristics. To accomplish this, we serially passaged the cultures and added 100 nM 4-hydroxy tamoxifen to the transwell culture media (Fig. 7a). The non-basal cell-derived and BASC-derived cultures continued robust growth in tamoxifen, but the AT2-derived cultures reproducibly failed to grow in tamoxifen (Fig. 7b and Supplementary Fig. 7a). Tamoxifen administration to AT2-derived cultures from WT mice continued to grow normally

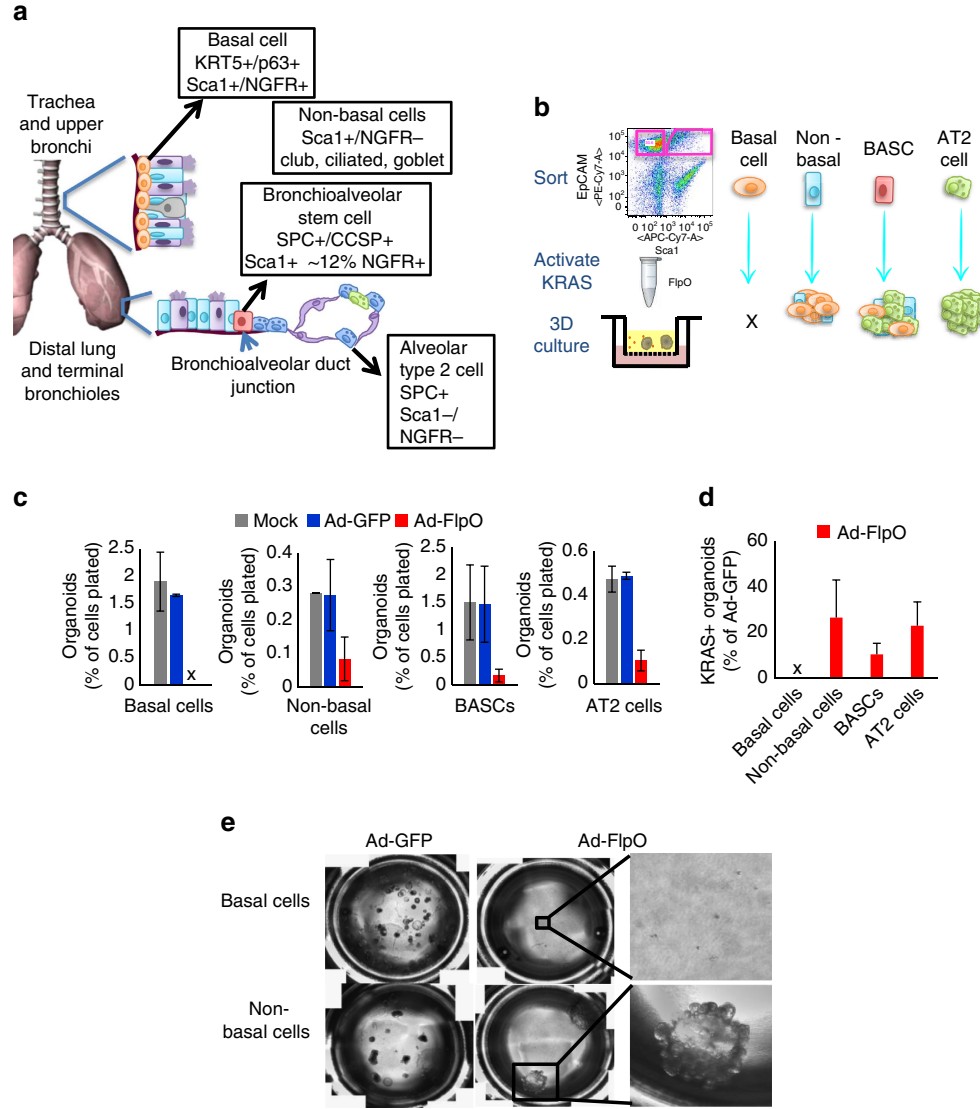

**Figure 6 | KRAS activation is tolerated by club cell, BASC and AT2 cell populations.** (**a**) Schematic of putative cells-of-origin of lung tumours, including basal tracheal cells, non-basal cells (enriched for club, ciliated and goblet cells), bronchioalveolar stem cells (BASCs) and alveolar type II (AT2) cells. (**b**) Schematic of sorting and *in vitro* activation of KRAS by FlpO prior to three-dimensional culture of lung cell populations described in (**a**). (**c**) Organoid growth from each of the lung stem cell populations mock transduced or transduced with adeno-GFP virus or adeno-FlpO virus. All are primary cultures quantified 9–14 days after seeding, $n=4$, mean ± s.e.m. is graphed, x indicates no growth. (**d**) KRAS+ organoid forming efficiency, graphed as the percentage of adeno-GFP control wells, $n=4$, mean ± s.e.m. is graphed. (**e**) Images of transwells seeded with basal cell or non-basal cells transduced with adeno-GFP or adeno-FlpO. See also Supplementary Fig. 6a–c.

(data not shown), suggesting that tamoxifen itself was not detrimental and that KRAS+ AT2 cells cannot tolerate loss of *Lkb1*. We confirmed *Lkb1* mRNA levels were decreased by tamoxifen administration in non-basal cell and BASC-derived cultures, and also that *Sox2* gene expression was increasing in these cultures, though Sox2 expression was variable (Fig. 7c). The variability of *Sox2* increase fit with the observation that only some organoids in each culture took on squamous characteristics (Supplementary Fig. 7b). Lastly, to demonstrate that we had taken a normal cell population and transformed it to a fully malignant state *in vitro*, we subcutaneously transplanted cells from the non-basal cell-derived cultures into immunocompromised mice. Large tumours formed (>500mm³), and upon histological examination, these tumours had regions of squamous differentiation right next to regions of ADC differentiation ($n=2$; Fig. 7d). We also transplanted cells orthotopically from

BASC-derived cultures, while one mouse had solely squamous differentiated tumour, the other had solely mucinous ADC ($n=2$, Fig. 7d and Supplementary Fig. 7c). Therefore, our experiments rule out the basal cells and AT2 cells as likely cells-of-origin for adenosquamous tumours, whereas both non-basal tracheal cells and distal lung BASCs could drive squamous disease.

## Discussion

Lung cancer lineage plasticity is an emerging concept that could influence the diagnosis and treatment of this devastating disease. In this manuscript, we used a unique model system to study lineage switching in KRAS-driven tumours in response to *Lkb1* deletion. To validate the clinical utility of our model, we first examined a cohort of ADSCC lung cancer, finding that

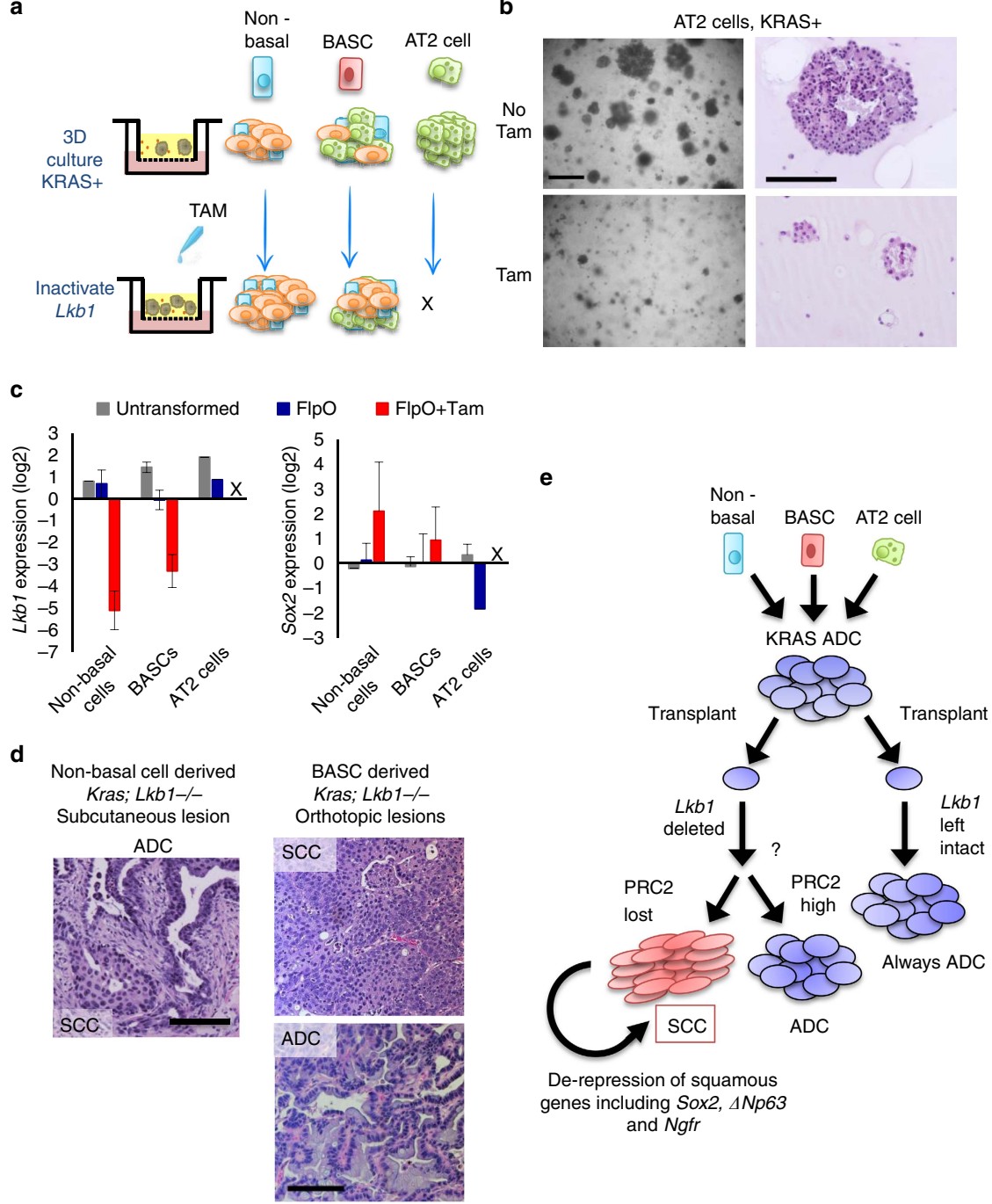

**Figure 7 | *Lkb1* deletion drives squamous transition of club cell- and BASC-derived KRAS+ populations.** (**a**) Schematic of *Lkb1* deletion in three-dimensional KRAS+ cultures derived from non-basal tracheal cells, bronchioalveolar stem cells (BASCs) or alveolar type II (AT2) cells. (**b**) Representative brightfield images of KRAS+ colonies from alveolar type II cells treated with placebo or 4-hydroxy tamoxifen for 7 days, scale bar, 200 μm. (**c**) RT-qPCR for *Lkb1* and *Sox2* in tumour organoid 3D cultures treated with 100 nM tamoxifen for 9–12 days, mean ± s.e.m. on log$_2$ scale is graphed, *n* varies by sample. (**d**) Representative haematoxylin and eosin staining from non-basal cell derived subcutaneous tumour (left) and BASC-derived orthotopic tumours (right), tumour histologies are indicated, scale bar, 100 μm. (**e**) Model: Here we combined Cre and FlpO recombinase technologies to temporally delete *Lkb1* in established KRAS-driven lung adenocarcinomas. Serially transplanted KRAS+ adenocarcinoma could transdifferentiate into squamous disease when *Lkb1* was deleted in the transplanted tumours. The Polycomb Repressive Complex 2 (PRC2), which represses genes through the histone H3K27me3 mark, was abrogated in the KRAS/*Lkb1* squamous tumours through loss of expression of the EED subunit. This led to derepression of key squamous genes including *Sox2*, *ΔNp63* and *Ngfr*. See also Supplementary Fig. 7a–c.

the *LKB1* mutation was present in more than one quarter of the patients. We next demonstrated that KRAS-driven ADCs, either in the autochthonous setting or after transplant, retained the ability to transition into squamous tumours when *Lkb1* was

deleted many weeks or months after KRAS induction. We further demonstrated that a loss of PRC2-mediated gene repression was a hallmark of the lineage switched SCC tumours. Lastly, our lung organoid data suggest that a combination genetic alterations,

epigenetic state and cell-of-origin ultimately determine tumour phenotype (Fig. 7e).

Our data indicate that after *Lkb1* is deleted in established KRAS ADCs, epigenetic reprogramming drives cells to take on squamous characteristics, ultimately resulting in full squamous transition of some tumours. These transitioned SCC tumours are characterized in both the mouse and human by a loss of the PRC2 H3K27me3 repressive chromatin mark, leading to activation of previously bivalently marked chromatin regions. Loss of PRC2-mediated gene repression leads to activation of squamous transcriptional programme, including the key squamous-associated genes *Ngfr*, *Sox2, ΔNp63* and *Krt5/6*. *Sox2* and *ΔNp63* are both well known to drive the squamous fate[44–47]. Thus, it is likely derepression of these genes that allows perpetuation of the squamous fate after the initial transition has occurred.

EZH2, which is the PRC2 methyltransferase that catalyses H3K27me3, has been proposed to be an oncogene in both lung ADC and SCC because its expression is correlated with poor prognosis of both histologies and its overexpression can drive ADC in mouse models[39,48–50]. Intriguingly, we and others have observed a decoupling of EZH2 expression levels from H3K27me3 levels in cancers[51–53]. In ADC regions, high and uniform H3K27me3 was found with relatively low expression of EZH2. It may be possible that EZH2 is only expressed in proliferating cells to replace the H3K27me3 marks lost during division. In SCC regions, there was often high EZH2, but lower levels of H3K27me3 mark than in ADC regions. We determined that reduction of H3K27me3 in SCCs was likely due to downregulation of the PRC2 component EED. EED loss may be the key to maintain an EZH2 high/H3K27me3 low state in other tumour types, or even in normal cells such as the oesophageal epithelium, and will warrant more study in the future. Because EZH2 was overexpressed in SCC tumours, it remained possible that EZH2 was critical for maintenance of the squamous fate with roles outside of the PRC2 complex, as has been observed in other tumour types[54–56]. However, genetic depletion of *Ezh2* actually potentiated KRAS/*Lkb1* SCC tumour growth, and EZH2 inhibition had no effect on KRAS/*Lkb1* 3D cultures. These data are supported by a recent study that demonstrated that *Eed* deletion accelerates KRAS/*p53* tumour development by driving a switch to mucinous ADC[39]. Both of these studies suggest that the PRC2 complex may act as a tumour suppressor in lung cancer by limiting lineage identity.

There has been much controversy over the possible cells-of-origin of subtype and genotype-specific lung tumours. After using intranasal adenovirus-Cre to induce oncogenic KRAS, the first hyperproliferative cells observed were BASCs, implicating this population as possible ADC cells-of-origin[19]. More recent studies with lineage-specific KRAS activation *in vivo* suggested that AT2 cells appeared to be the only cells capable of giving rise to advanced ADC in the alveolar space, while club cells and BASCs appeared to be limited to driving bronchiolar hyperplasia within the same time frame[57]. Importantly, results varied when the genotype was altered to include SOX2 activation or p53 loss[58,59]. Our data add to this picture by indicating that NGFR+ basal cells cannot tolerate oncogenic KRAS. Given that basal cells express high levels of EGFR[37], and that KRAS activation and EGFR activation cannot be tolerated by the same cell[60], it is conceivable that basal cells undergo oncogene-induced senescence upon KRAS activation. In the more distal lung, we observed that both AT2 cells and BASCs could efficiently form KRAS+ organoids. However, the AT2-derived cultures could not proliferate after *Lkb1* deletion. It was the non-basal cells of the trachea, a population enriched for club, ciliated and goblet cells[23,43], and the BASCs of the

distal lung that could survive both KRAS activation and *Lkb1* deletion. Both cell types gave rise to organoids that could grow *in vivo*, with some ADC and some SCC histology. These findings are in strong agreement with a recent paper using cell type restricted Cre virus, which also found the club cells to be more likely cells-of-origin of adenosquamous tumours of the KRAS/*Lkb1* genotype[36]. These results are consistent with recent evidence that both CCSP and SPC expressing cells can give rise to SCCs with activation of SOX2 and inactivation of *Pten* and *Cdkn2a*[45]. Overall our data suggest that club cells and BASCs are the most likely cells of origin of KRAS/*Lkb1* ADSCC, and that further characterization of these cell types in their normal state and in the context of oncogenic activation will shed new light on the subtype of adenosquamous cancer in patients.

What remains unclear is what why certain cells respond to *Lkb1* deletion by downregulation EED and ultimate loss of PRC2-mediated gene repression of the squamous transcriptional programme. Because LKB1 affects many metabolic processes, one possibility is that metabolism of methionine is a link between LKB1 and EED. Studies with an inhibitor of *S*-adenosyl homocysteine hydrolase have demonstrated that decreased methionine metabolism can cause destabilization of the PRC2 components at the protein level[61]. Future studies will focus on identifying the link between the cellular genotype and the epigenetic identity of lung cancer cells from different subtypes of lung cancer, including ADC, SCC and ADSCC, with the ultimate goal of identifying unique vulnerabilities of each that can be used in new therapeutic approaches.

## Methods

**Mouse cohorts.** Mouse cohorts of Lkb1[flox/flox]/Pten[flox/flox]; LSL:Kras[G12D/+], LSL:Kras [G12D/+]; Lkb1[flox/flox], and FSF:Kras[G12D/+]; FSF:R26:CreERT2; Lkb1[flox/flox] and LSL:Kras [G12D/+]; Lkb1[flox/flox], Ezh2[floxed] were all maintained in virus-free conditions on a mixed 129/FVB background. FSF:Kras[G12D] (Krastm[5Tyj]) mice were purchased from The Jackson Laboratory; Foxn1/Foxn1[Nu/Nu] mice were purchased from Charles River Laboratories International Inc. All care and treatment of experimental animals were in strict accordance with Good Animal Practice as defined by the US Office of Laboratory Animal Welfare and approved by the Dana-Farber Cancer Institute Institutional Animal Care and Use Committee. Mice were given $2 \times 10^8$ pfu of Adeno-FlpO virus via intranasal infections at 6–8 weeks old. Mice were administrated daily by intraperitoneal injection of tamoxifen-free base (SIGMA T5648) in corn oil (12 μg ml$^{-1}$) at 132 μg tamoxifen per gram of body weight per day for five consecutive days. Magnetic resonance imaging scan was used for mouse lung tumour burden measurement as previously described[37]. Intratracheal transplants were performed as described[62]. Mice were monitored for signs of lung tumour onset and killed for gross and histological analysis and tumour isolation upon signs of distress.

**Flow cytometry analysis and sorting.** Tumours were dissected from the lungs of primary mice and tumour tissue was prepared as described[62]. Single-cell suspensions were stained using rat-anti-mouse antibodies. Antibodies for tumour cell analysis included anti-human-NGFR (EMD Millipore 05-446) coupled with goat-anti-mouse-PECy7 (BioLegend 405315), anti-mouse-NGFR (AbCAM ab8875) coupled with donkey-anti-rabbit-PE (eBiosciences 12-4739-81), anti-mouse-Sca1-APCCy7 (Fisher Scientific BDB560654), anti-mouse-EpCAMPECy7 (BioLegend 118216), anti-mouse-CD31-APC (Fisher Scientific BDB551262) and anti-mouse-CD45-APC (Fisher Scientific BDB559864). Live cells were gated by exclusion of 40,6-diamidino-2-phenylindole (DAPI)-positive cells (SIGMA). Immune cell analysis is described in a previous study[63]. All antibodies were incubated for 10-15 min at 1:100 dilutions for primaries and 1:200 for secondary antibodies. Cell sorting was performed with a BD FACS Aria II with an 85 mm nozzle, and flow cytometric analysis was performed with a BD Fortessa and data were analysed with FlowJo software (Tree Star).

***In vitro* FlpO and stem cell organoid culture.** Cells isolated by flow cytometry as described above were split into equal aliquots, pelleted by pulse spin and resuspended in 100 μl MTEC plus media with $6 \times 10^7$ PFU per ml$^{-1}$ of Adeno-FlpO or Adeno-GFP or no virus. Cell numbers for each condition/well ranged from 1,000–10,000 basal cells, 2,000–10,000 non-basal cells, 33,000–100,000 AT2 cells and 2,000–20,000 BASCs. These suspensions were incubated at 37C, 5% $CO_2$ for 2 h in 1.5 ml tubes. Mock-infected cells (no virus) were also incubated

for the same time to control for anoikis. Cells were then pelleted by pulse spin and resuspended in $1\times$ phosphate-buffered saline (PBS) twice to wash and finally resuspended for 3D culture. Tracheal cells were cultured as described in ref. 12, with additional $25\,ng\,ml^{-1}$ rmFGF2 (R + D Systems 3139-FB/CF) and in a total starting volume of $100\,\mu l$ of cells in media mixed with $100\,\mu l$ of growth factor reduced Matrigel (Fisher Scientific). Previously it was demonstrated that distal lung stem cells can differentiate into lineage-specific organoids when grown in cultures with primary lung endothelial cells (see ref. 20), and we employed these cultures for NGFR+ and NGFR- BASCs as well as AT2 cells. Tracheal cells grown in the same endothelial cell cultures grew similarly hollow spheres composed of basal-like p63+ cells, what appear to be lineage-negative cells and ciliated cells, just as they do in the MTEC media.

**Immunofluorescence on organoid cultures.** Cultured colonies were fixed with 10% neutral-buffered formalin in overnight at room temperature. After rinsing with 70% ethanol, fixed colonies were immobilized with Histogel (Thermo Scientific) for paraffin embedding. Sectioned embedded colonies were stained with haematoxylin and eosin and immunostained with antibodies for CCSP (1:200, Santa Cruz clone T-18), pro-SPC (1:200, Santa Cruz clone FL-197), p63 (1:200, Biocare clone 4A4), acetylated-tubulin (1:1,000; Sigma, clone 6-11B-1). Secondary antibody staining included donkey a-mouse 488, donkey a-rabbit 594, donkey a-goat 647 (1:400, Invitrogen) and Prolong Gold with DAPI (Life Technologies).

**Tumour cell 3D culture.** Dissociated tumour cells were resuspended in DME/F12 media (GIBCO) containing $1\times$ penicillin/streptomycin (Invitrogen), 4 mM L-glutamine (Invitrogen), 10% fetal bovine serum (HyClone), $10\,\mu g\,ml^{-1}$ insulin (SIGMA I-6634), $1\times$ insulin/transferrin/selenium mixture (Corning 25-800-CR), $12.5\,\mu g\,ml^{-1}$ bovine pituitary extract (Invitrogen 13028-014), $0.1\,\mu g\,ml^{-1}$ cholera toxin (SIGMA C-8052), $25\,ng\,ml^{-1}$ mEGF (Invitrogen 53003018) and $25\,ng\,ml^{-1}$ rmFGF2 (R + D Systems 3139-FB/CF), mixed 1:1 with growth factor-reduced Matrigel (Fisher Scientific), and pipetted into a 12-well $0.4\,\mu m$ Transwell insert (Falcon). MTEC/Plus (described above) medium ($500\,\mu l$) was added to the lower chamber and refreshed every other day. GSK126 was purchased from Xcess Bio as a 10 mM solution in DMSO, and 4-hydroxy-Tamoxifen (SIGMA) was resuspended to a concentration of $100\,\mu M$ in sterile 100% ethanol. Both were diluted 1:1,000–1:2,000 into tissue culture media for use.

**Quantitative RT PCR.** RNA from treated cell lines was extracted using Absolutely RNA kits (Agilent) and cDNA was made using the SuperScript III kit (Invitrogen). Relative gene expression was assayed with Taqman probes on the StepOnePlus Real-Time PCR System (Applied Biosystems). Relative expression was calculated by Gene of Interest $(Ct_{reference} - Ct_{expreimental}) - CYPA(Ct_{reference} - Ct_{experimental})$ and graphed on the $\log_2$ scale or converted to linear scale. Statistics were performed on $\log_2$ data. For all experiments, the reference sample was a matched vehicle treated or control transduced cell line.

**Statistical analysis.** Statistical analyses were carried out using GraphPad Prism. All numerical data are presented as mean ± s.e.m. Grouped analysis was performed using two-way ANOVA. Column analysis was using one-way ANOVA or $t$-test. A $P$-value less than 0.05 was considered statistically significant.

**Histology and immunohistochemistry.** Mice were killed with $CO_2$ and the right lobe was dissected and snap-frozen for biochemical analysis. The remainder of the lungs was inflated with neutral-buffered 10% formalin overnight at room temperature, and then transferred to 70% ethanol, embedded in paraffin and sectioned at $5\,\mu m$. Haematoxylin and eosin stains were performed in the Department of Pathology in Brigham and Women's Hospital. Tumour burden analysis was performed with ImageJ software. Immunohistochemistry was performed as previously described[37]. Antibodies used for other markers are listed below: LKB1 (Cell Signaling 3047S), TTF1 (DAKO M3575), p63 (AbCAM ab53039), SOX2 (Millipore AB5603), NGFR (Epitomics 1812-1), MPO (Novus R-1073), F4/80 (eBioscience 14-4801-82) and p63 (Santa Cruz sc-25268). Scoring of H3K27me3 (Cell Signaling 9733) and EZH2 (Cell signaling 5246S) nuclear staining was performed with Nikon NIS-Elements AR software. Magnification images of $\times 20$ or $\times 40$ were captured of both ADC and SCC regions as judged by a pathologist. Regions of stroma (blood cells, vessels, fibroblasts and connective tissue) were deleted from the images by hand, prior to using the dot detection module to quantify all nuclei on the red channel, and to quantify stained nuclei with increasing stringency on the blue channel. Visual confirmation of one dot per nucleus and cutoff for stained signal were confirmed by the user for each set of images. Importantly, cutoff settings remained the same for ADC and SCC images from the same slide, allowing direct comparison of staining intensities in the mixed adenosquamous cases. Percentages of positive nuclei were quantified and graphed.

**Western blot.** Whole-cell extracts were made in RIPA buffer (0.5% deoxycholate, 1% IGEPAL-CA630, 0.1% sodium dodecyl sulfate, 150 mM NaCl, 50 mM Tris-8.1), lysates were cleared by centrifugation and protein concentrations were quantified with the Pierce BCA Protein Assay Kit (Thermo). For western blotting, $25\,\mu g$ of protein extract per sample was denatured with heat and reducing agents, separated on a 4–12% acrylamide gel (BioRad) and transferred to nitrocellulose (GE Healthcare). Antibodies used for western blotting were EZH2 (BD Transduction Laboratories 612666, 1:1,000), LKB1 (Cell Signaling 3047S, 1:1,000), SUZ12 (AbCAM ab12073, 1:1,000), EED (Millipore 09-774, 1:500), JMJD3 (Aviva Systems Biology ARP40101_P050, 1:500), UTX (GeneTex GTX121246, 1:1,000), total Erk1,2 (Cell Signaling Technology 4695S 1:1,000), pERK1,2 (Cell Signaling Technology 4376S, 1:1,000), pAMPK (Thr172) (40H9 Cell Signaling 2535 1:1,000), Histone H3 (AbCAM ab1791, 1:20,000), H3K27me1 (Millipore 07-448 1:2,000), H3K27me2 (Millipore 07-452 1:2,000) and H3K27me3 (Millipore 07-449; 1:4,000) all incubated overnight at $4\,°C$. β-Actin-HRP (SIGMA A3854, 1:20,000) was used as a loading control. All antibodies have detailed species validation available online from vendors. Secondary antibody Anti-rabbit IgG, HRP-linked Antibody (Cell Signaling 7074, 1:2,000) or Anti-mouse IgG, HRP-linked Antibody (Cell Signaling 7076, 1:2,000) was incubated for 1 h at room temperature. After washing, chemiluminescence was visualized with Western Lightning Plus-ECL (PerkinElmer) and exposed onto KODAK BioMax XAR film.

**Chromatin immunoprecipitation and sequencing (ChIP-seq).** Lung tumours were pulverized, crosslinked with 1% formaldehyde in PBS for 10 min at RT, washed in $5\,mg\,ml^{-1}$ bovine serum albumin in PBS and then in just cold PBS, re-suspended in lysis buffer (50 mM Tris-HCl, pH 8.1, 10 mM EDTA, 1% SDS, $1\times$ complete protease inhibitors (Roche)) and sonicated with the Covaris E210 or S2 sonicator to obtain chromatin fragment lengths of 200-to-1,500 bp judged by Bioanalyzer DNA High sensitivity kit (Agilent). Fragmented chromatin was diluted in IP buffer (20 mM Tris-HCl pH 8.1, 150 mM NaCl, 2 mM EDTA, 1% Triton X-100) and incubated overnight at $4\,°C$ with Protein G magnetic beads (Dynabeads: Life Technologies) that had been pre-incubated with H3K4me3 (Millipore 07-473), H3K27Ac (Abcam ab4729) or H3K27me3 (Cell Signaling Technologies C36B11) antibodies. Immunoprecipitates were washed six times with wash buffer (50 mM HEPES pH 7.6, 0.5 M LiCl, 1 mM EDTA, 0.7% Na deoxycholate, 1% NP-40) and twice with TE buffer. Immunoprecipitated (or no IP input) DNA was treated with RNase A and Proteinase K on the beads, recovered in 1% SDS and 0.1 M NaHCO$_3$ over a period of 5 h at $65\,°C$, column purified with DNA clean and concentrator-25 (Zymo Research). After a sonication to enrich DNA fragment lengths between 100 and 300 bp, 5–10 ng of DNA were used for the library construction using NEBNext Ultra DNA Library Prep Kit (NEB E7370). Sequencing was performed on a NextSeq (Illumina) for 38 nucleotides from paired ends according to the manufacturer's instructions.

**ChIP-seq analysis.** Sequenced reads were mapped to reference mouse genome build 38 (mm10), using bowtie2 aligner[64]. H3K4me3, H3K27Ac modified regions were identified using MACS (version 1.4.2) as previously described (Zhang et al., 2008)[65], with a $P$-value cutoff of $10^{-5}$ and with default values for other parameters. Computer code is available upon request. Quantitative differences of histone modification between ADC and SCC tumours from Kras/Lkb1 mice were analysed using the MAnorm algorithm[66] with a $P$-value cutoff of $10^{-10}$ and $> 1\,\log_2$ fold change for H3K4me3 and H3K27Ac.

Wiggle files with a 10 bp resolution for H3K4me3, H3K27ac and H3K27me3 modifications were generated by MACS v1.4.2 with tag shift and then rescaled to a larger of total number of uniquely alignable sequences in two data sets. Histone modification profiles were then generated with the 'heatmap' function or the SitePro module in Cistrome analysis pipeline (http://cistrome.org/). Wiggle files were also visualized in the Integrative Genomics Viewer (Robinson et al., 2011)[67].

To correlate ChIP-Seq and gene set enrichment, peaks were associated with genes using GREAT[42], allowing for a maximum distance of 10 kb between peak and associated gene, irrespective of directionality. Data are available at GEO repository (accession number GSE94365).

**Patient lung tumours.** Between July 2013 and April 2016, 9616 patients underwent targeted next-generation sequencing (NGS) at the Dana-Farber Cancer Institute/Brigham and Women's Hospital with informed consent under an IRB-approved research protocol. Among these population, 14 patients with adenosquamous non-small-cell lung cancer and 84 patients with an LKB1 mutation and ADC were identified using the Oncology Data Retrieval System (OncDRS). OncDRS is an internal system developed at the Dana-Farber Cancer Institute to integrate clinical and genomic data. All patients provided written informed consent, and no germline sequencing was performed. Tumour cell sequencing was performed on tumour DNA extracted from fresh, frozen or formalin-fixed paraffin-embedded samples and evaluated for single-nucleotide variants and genomic rearrangements[68]. The initial gene panel surveyed all exons of the 275 genes in the panel, along with 91 introns across 30 genes for rearrangement detection. DNA was isolated from tissue containing at least 20% tumour nuclei and analysed by massively parallel sequencing using a solution-phase Agilent SureSelect

hybrid capture kit and an Illumina HiSeq 2500 sequencer. Data were analysed by an internally developed bioinformatics pipeline composed of reconfigured publicly available tools and internally developed algorithms including Indelocator http://www.broadinstitute.org/cancer/cga/indelocator and Oncotator[69,70]. Samples with a mean target coverage of $<50\times$ were excluded from further analysis. Individual variants present at $<10\%$ allele fraction or in regions with $<50\times$ coverage were flagged for manual review and interpreted by the reviewing laboratory scientists and molecular pathologists based on overall tumour percentage, read depth, complexity of alteration and evidence for associated copy number alterations.

**Data availability.** Supporting data are available upon request from the corresponding authors.

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

## Acknowledgements

We thank the Kim and Wong Labs for helpful discussions. This work was supported in part by PF-12-151-01-DMC from the American Cancer Society, the Uniting Against Lung Cancer Young Investigator Award and NCI K22 CA201036 (to C.F.B.), R01 HL090136, R01 HL132266, R01 HL125821, U01 HL100402 RFA-HL-09-004, American Cancer Society Research Scholar Grant RSG-08-082-01-MGO, the V Foundation for Cancer Research, a Basil O'Conner March of Dimes Starter Award and the Harvard Stem Cell Institute (to C.F.K.), the NIH/NCI P01 CA120964, 5R01CA163896-04, 1R01CA195740-01, 5R01CA140594-07, 5R01CA122794-10 and 5R01CA166480-04 grants and Support from Gross-Loh Family Fund for Lung Cancer Research and Susan Spooner Family Lung Cancer Research Fund at Dana-Farber Cancer Institute (to K.K.W.).

## Author contributions

H.Z., C.F.B., S.K., H.W., C.F.K. and K.K.W conceived the ideas and designed the experiments, acquired the data and performed the analyses and interpretations. M.G., C.G.A., M.P., G.S.H, G.L., X.Z., B.P.M., S.J.T., C.X., Z.C., X.W. and E.A. provided technical assistance. T.C. and S.L. assisted with mouse experiments. A.J.R., L.M.S. and M.Z. provided clinical samples and contributed to clinical data analyses. S.P., A.K.R., D.J.K, J.A.D., A.J.B., N.E.S., G.D., P.S.H., H.J., N.B. and D.S. provided materials and contributed to data interpretation and manuscript preparations. L.M.S. contributed to pathological examination. H.Z., C.F.B., C.F.K. and K.K.W. wrote the manuscript.

## Additional information

**Competing interests:** The authors declare no competing financial interests.

DOI: 10.1038/ncomms15901 OPEN

# Erratum: *Lkb1* inactivation drives lung cancer lineage switching governed by Polycomb Repressive Complex 2

Haikuo Zhang, Christine Fillmore Brainson, Shohei Koyama, Amanda J. Redig, Ting Chen, Shuai Li, Manav Gupta, Carolina Garcia-de-Alba, Margherita Paschini, Grit S. Herter-Sprie, Gang Lu, Xin Zhang, Bryan P. Marsh, Stephanie J. Tuminello, Chunxiao Xu, Zhao Chen, Xiaoen Wang, Esra A. Akbay, Mei Zheng, Sangeetha Palakurthi, Lynette M. Sholl, Anil K. Rustgi, David J. Kwiatkowski, J. Alan Diehl, Adam J. Bass, Norman E. Sharpless, Glenn Dranoff, Peter S. Hammerman, Hongbin Ji, Nabeel Bardeesy, Dieter Saur, Hideo Watanabe, Carla F. Kim & Kwok-Kin Wong

*Nature Communications* 8:14922 doi: 10.1038/ncomms14922 (2017); Published 7 Apr 2017; Updated 9 Jun 2017

The affiliation details for Hideo Watanabe are incorrect in this Article. The correct affiliation details for this author are given below:

Department of Medicine, Division of Pulmonary, Critical Care and Sleep Medicine; Tisch Cancer Institute, Icahn School of Medicine at Mount Sinai, New York, New York 10029, USA.

