## [Peer Review File · Nature Communications]

Reviewers' comments:

Reviewer #1 (Remarks to the Author):

In recent years, there has been increasing evidence of tumor cell plasticity. This can have significant consequences therapeutically and therefore it is very important to understand the molecular mechanisms that govern phenotypic changes in tumor cells. Therefore, efforts like those described in this paper are important. Here, the authors investigate the mechanisms that govern the transdifferentiation of lung adenocarcinomas (ADC) to squamous cell carcinoma (SCC) in mouse models of Kras/Lkb lung cancer. The finding that this transdifferentiation occurs has previously been published (PMID: 24531128). However, the transplantation experiment in Figure 2 is further evidence of transdifferentiation occurring and is a useful addition to the field. The authors make the novel observation that changes in expression of PRC2 complex components are altered in SCCs and that EZH2 loss in this context exacerbates the transdifferentiation of ADCs to SCCs as a result of chromatin changes that lead to expression of squamous differentiation genes. The authors also identify a possible relationship between genes regulated by PRC2 in the SCCs and neutrophil infiltration in the tumors. This unfortunately is analyzed only superficially but could potentially represent an important aspect of the transdifferentiation/squamous differentiation process (are neutrophils important for the transdifferentiation?). Finally, the study investigates the properties of different cell populations in the lungs of these models. Unfortunately, this part of the manuscript is very confusing and lacks a clear description of why the experiments were performed and what is learned from the studies. The results in Figures 6 and 7 therefore should be significantly edited so that the reasoning and findings can be better interpreted.

In summary, this paper sheds light on an important process with several novel findings although it requires the following significant revisions:

- 1) since the authors barely touch on the topic of neutrophils and only show correlative data they should remove this altogether or add experiments to describe the role of these cells in the transdiff/squamous cell diff process.
- 2) Data in figures 6+7 on the cell of origin should be streamlined and described better so that a reviewer can evaluate its significance.

Minor comments:

- 1) the word "supplementary" randomly is found in the text when it shouldn't be present.
- 2) page 7. It isn't clear which experiment was done on a single mouse. Since there is a survival curve, the authors have analyzed several mice. They should clarify.
- 3) what is the status of the PRC2 complex in LKB/PTEN SCC tumors? This information would be helpful in the context of their experiments since it seems like disruptions in PRC2 potentiate SCC differentiation that is very abundant in the LKB/PTEN system.
- 4) Data in ref 39 in their paper and their data suggest that LKB1 is still very important for the SCC diff process to occur. The authors should discuss why and what it may be doing.
- 5) Page 8: Figure 4d. The authors jump to this experiment but it is disconnected from the rest. A better transition should be added.
- 6) Figure 1a: the schema should be clearer since it is hard to understand when tamoxifen is given and cre induced.
- 7) TAM is used for tamoxifen and for macrophages in the text. It would be good to just use if for one of these to avoid confusion.

Reviewer #2 (Remarks to the Author):

The authors previously demonstrated a mixed ADC, SCC tumor phenotype in KRas/Lkb1 mutant NSCLC models. They now convincingly show by sequential inactivation of Lkb1 in the KRAs mutant model that loss of Lkb1 promotes the transition from DC to SCC. They observe that the H3K27me3

Polycom repressive mark is lowered in the SCCs, presumably by loss at the protein level of the essential PRC2 subunit Eed (while curiously the catalytic subunit levels are increased- in most other described cases loss of Eed leads to instability of PRC2 including Ezh2. The transition to SCC is at the marker level not complete, as some TTF1 is retained in the SCCs they observe. They further show using organoid cultures that also enzymatic inhibition of Ezh2 results in a partial transition towards SCC, suggesting that it is indeed PRC2 functional loss that promotes the epigenetic transition. This is also confirmed using ChIPseq where they see a transition to active marks at genes (and putative enhancers) that would normally be silenced by PRC2. These do include important markers for SCC such as Sox2 and Ngfr. In additional sets of experiments where they enrich for certain subsets of lung progenitors using FACS markers they demonstrate differential effects of growth promoting or contrary growth arresting effects of the different lesions and also of clear effects on the tumor microenvironment, especially the attraction of neutrophils who could have important tumor promoting effects and assist in the SCC transition. Arguably, while the effects in the mouse models are clear, the effects of lowering H3K27me3 in the few human cases shown is much weaker, which may limit the extrapolation and importance for human SCC in NSCLC. This is also of importance with regard to the fact that Lkb1 deletion in KRAs mutant human NSCLC is rare. It also remains unclear why Lkb1 loss promotes (in a subset of tumors) the loss of Eed protein- here more mechanistic insight is required, and indeed one might ask why they did not validate their observation by using Eed conditional mice instead of Ezh2 floxed mice. In addition, it remains unclear which of the derepressed Polycom targets are essential for the observed effects towards SCC differentiation. Notwithstanding these points, it is of interest that now in the Lkb1/Kras mutant setting also differential effects are seen for loss of PRC2 function, with some similarities to a recent study documenting a pro-oncogenic effect of PRC2 in KRas mutant setting to a tumor suppressive effect in the KRas/P53 mutant setting, also involving strongly the tumor microenvironment.

Letter to Reviewers:

Thank you both very much for the critical reading of our manuscript, and for suggested edits. Below is a point-by-point response to each of your remarks.

Reviewer #1 (Remarks to the Author):

Reviewer: In recent years, there has been increasing evidence of tumor cell plasticity. This can have significant consequences therapeutically and therefore it is very important to understand the molecular mechanisms that govern phenotypic changes in tumor cells. Therefore, efforts like those described in this paper are important. Here, the authors investigate the mechanisms that govern the transdifferentiation of lung adenocarcinomas (ADC) to squamous cell carcinoma (SCC) in mouse models of *Kras/Lkb* lung cancer. The finding that this transdifferentiation occurs has previously been published (PMID: 24531128). However, the transplantation experiment in Figure 2 is further evidence of transdifferentiation occurring and is a useful addition to the field.

Response: Thank you for this supportive view of our addition to this field. We agree that the transplantation of the stepwise model cells are key pieces of data for our manuscript, as they definitively demonstrate that cells capable of serially passaging adenocarcinomas can become squamous in differentiation state after *Lkb1* deletion. We decided on the terminology of transition rather than transdifferentiation because in the cancer setting, the reprogramming of cells from one lineage to another cannot be decoupled from proliferation of the cells.

Reviewer: The authors make the novel observation that changes in expression of PRC2 complex components are altered in SCCs and that EZH2 loss in this context exacerbates the transdifferentiation of ADCs to SCCs as a result of chromatin changes that lead to expression of squamous differentiation genes. The authors also identify a possible relationship between genes regulated by PRC2 in the SCCs and neutrophil infiltration in the tumors. This unfortunately is analyzed only superficially but could potentially represent an important aspect of the transdifferentiation/squamous differentiation process (are neutrophils important for the transdifferentiation?).

Response: Thank you for recognizing our observation about the PRC2 complex in the transitioned tumors. In our new manuscript, we have explored this relationship of high EZH2 and low H3K27me3 in many more human tumor samples. Together with a published manuscript¹, it appears that high EZH2 and low H3K27me3 is a phenotype of many squamous tumors. In adenosquamous patients, who we argue are the patients we are modeling due to similarity in the genetics of these mice and human disease, we found that EZH2 was higher and H3K27me3 levels lower in the SCC portions of the tumors compared to ADC regions on the same tissue slides.

For the neutrophil signature, we have taken your suggestion to limit mention of these results as we had only performed one preliminary functional study. Neutrophils are very difficult to target *in vivo* because of their vast number and quick turnover. Similarly, *in vitro*, neutrophils only survive a maximum of 24 hours. These two points limited the experiments we could perform in the timeframe of a revision. Future studies could use genetic depletion of neutrophil lineages in the tumors to definitely study their contribution to the squamous transition.

Finally, the study investigates the properties of different cell populations in the lungs of these models. Unfortunately, this part of the manuscript is very confusing and lacks a clear description of why the

experiments were performed and what is learned from the studies. The results in Figures 6 and 7 therefore should be significantly edited so that the reasoning and findings can be better interpreted.

Response: Thank you for this valuable suggestion. We reformatted Figures 6 and 7. We added some new data of these cell populations growing as allografts in immunocompromised mice, demonstrating that we have taken a normal lung cell population and transformed it to a fully malignant cell population *in vivo*. Our results show that non-basal cells (and most likely the club cells within the non-basal fraction) or BASCs can act as cells-of-origin for KRAS-driven adenosquamous tumors.

In summary, this paper sheds light on an important process with several novel findings although it requires the following significant revisions:

1) Since the authors barely touch on the topic of neutrophils and only show correlative data they should remove this altogether or add experiments to describe the role of these cells in the transdiff/squamous cell diff process.

We agree. In the revised manuscript, we only left the supplemental immunostaining of the ADC and SCC lesions to show the preference of neutrophils in the squamous lesions. We also left the ChIP-seq peaks for *Cxcl3* and *Cxcl5*, which may be the mechanism through which the neutrophils are recruited. The initial studies are in line with our previous report of neutrophils in squamous tumors of the *Lkb1/Pten* genotype² and more studies can be done in the future, as the reviewer suggests.

2) Data in figures 6+7 on the cell of origin should be streamlined and described better so that a reviewer can evaluate its significance.

We have significantly revised these Figures with new data, including results of new transplantation experiments. Given the ability of adenocarcinoma cells to transition to squamous lesions, we wished to determine which lung cell types might be able to give rise to adenosquamous tumors in the context of KRAS activation and *Lkb1* loss. We hypothesize that by uncovering the cells of origin of lung tumors, that we can better understand the potential vulnerabilities of subtypes of lung cancer. We compared the ability of several different normal lung cell populations to grow as organoids: tracheal basal cells, non-basal tracheal cells, BASCs and alveolar type II cells. Our data suggest that amongst these cellular populations, club cells or BASCs can give rise to adenosquamous tumors with the combination of KRAS activation and *Lkb1* loss.

Minor comments:

1) the word "supplementary" randomly is found in the text when it shouldn't be present.

Corrected.

2) page 7. It isn't clear which experiment was done on a single mouse. Since there is a survival curve, the authors have analyzed several mice. They should clarify.

We apologize for the confusing statement. We did perform the survival curve by analyzing many mice, and indicated n on the graph. Likewise, we did the histology analysis on many mice, again with the n indicated on the graphs. What we meant by 'in a single mouse' was that, in our first set of experiments, even though KRAS was activated at a different time than *Lkb1* was deleted, the tumors remained within the same mouse during the whole of the tumor development. In contrast, in the second set of experiments, we transplanted the KRAS adenocarcinomas prior to *Lkb1* deletion. We rewrote this section to clarify.

3) what is the status of the PRC2 complex in LKB/PTEN SCC tumors? This information would be helpful in the context of their experiments since it seems like disruptions in PRC2 potentiate SCC differentiation that is very abundant in the LKB/PTEN system.

To address this comment, we stained for EZH2 and H3K27me3 on *Lkb1/Pten* squamous, *Lkb1/Pten/p53* adeno-squamous tumor slides. We observed a similar pattern of staining, with many regions of squamous histology exhibiting higher EZH2 and lower H3K27me3 than the adenocarcinoma regions. These new data are provided in Figure 3 and Supplementary Figure 3. These results mirror our results with the human patient samples: in 6 patients with adeno-squamous disease we compared EZH2 and H3K27me3 staining intensities in regions of both histologies in the same tissue sections. Consistently, adenocarcinoma cells had very high H3K27me3 levels while squamous regions had lower, yet not absent, H3K27me3 levels. In contrast, adenocarcinoma regions had low to absent EZH2 levels, with rare cells expressing high levels. Squamous tumors often had regions of many cells with high EZH2 levels, though they could still have regions that were low to negative for the marker, suggesting that EZH2 is dispensable for maintenance of the squamous state.

4) Data in ref 39 in their paper and their data suggest that LKB1 is still very important for the SCC diff process to occur. The authors should discuss why and what it may be doing.

This is an excellent point. Yes, we completely agree that LKB1 depletion is required for the transition in mice, and the human data demonstrating that *LKB1* mutation in adenosquamous tumors is very common further support the connection. Control of cellular methionine metabolism by LKB1 is now one of our main hypotheses. Specifically, we believe that depletion of LKB1 may alter cellular metabolism such that cells may be caused to decrease methionine metabolism in order to maintain proliferation. Future studies will test this hypothesis. We have added this point to the discussion.

5) Page 8: Figure 4d. The authors jump to this experiment but it is disconnected from the rest. A better transition should be added.

Thank you for this comment, we added a full transition sentence to better link the *Ezh2* floxed mouse model to EZH2 inhibition in culture.

6) Figure 1a: the schema should be clearer since it is hard to understand when tamoxifen is given and cre induced.

We revised this figure and hope that the reviewer finds it easier to understand.

7) TAM is used for tamoxifen and for macrophages in the text. It would be good to just use if for one of these to avoid confusion.

Thank you for this helpful suggestion. We retained the use of TAM only for tamoxifen, and used the full name tumor-associated macrophages for the supplemental figure showing immunostaining.

Reviewer #2 (Remarks to the Author):

The authors previously demonstrated a mixed ADC, SCC tumor phenotype in KRas/Lkb1 mutant NSCLC models. They now convincingly show by sequential inactivation of Lkb1 in the KRAs mutant model that loss of Lkb1 promotes the transition from ADC to SCC. They observe that the H3K27me3 Polycomb repressive mark is lowered in the SCCs, presumably by loss at the protein level of the essential PRC2 subunit Eed (while curiously the catalytic subunit levels are increased- in most other described cases loss of Eed leads to instability of PRC2 including Ezh2).

Response: Thank you for these endorsements of our experimental methods. We would like to address the catalytic function of EZH2 here: In the literature, there are several tumor types where EZH2 and H3K27me3 have inverse staining patterns, including in lung SCC¹, breast cancer³ and pancreatic cancer⁴. While these manuscripts did not analyze EED levels (arguably there is no good EED IHC antibody for human), it remains very likely that loss of EED and/or SUZ12 expression in these tumor types is the mechanism through which an EZH2 high/H3K27me3 low state is attained and maintained. What the residual EZH2 protein is doing, if not catalyzing H3K27me3, is unknown. There are hypotheses that EZH2, outside of the PRC2 complex, could act as a transcriptional activator⁵⁻⁷. We demonstrate that EZH2 itself is dispensable for the squamous phenotype with our genetic approach. Therefore, even if EZH2 is acting separately from the PRC2 complex as a transcriptional activator in the squamous tumor cells, this function is not required. In contrast, depletion of H3K27me3 in KRAS/Lkb1 adenocarcinomas is sufficient to drive squamous differentiation, suggesting loss of PRC2-mediated gene repression is one mechanism through which adenosquamous tumors arise in patients.

The transition to SCC is at the marker level not complete, as some TTF1 is retained in the SCCs they observe. They further show using organoid cultures that also enzymatic inhibition of Ezh2 results in a partial transition towards SCC, suggesting that it is indeed PRC2 functional loss that promotes the epigenetic transition. This is also confirmed using ChIPseq where they see a transition to active marks at genes (and putative enhancers) that would normally be silenced by PRC2. These do include important markers for SCC such as Sox2 and Ngfr. In additional sets of experiments where they enrich for certain subsets of lung progenitors using FACS markers they demonstrate differential effects of growth promoting or contrary growth arresting effects of the different lesions and also of clear effects on the tumor microenvironment, especially the attraction of neutrophils who could have important tumor promoting effects and assist in the SCC transition.

Response: Thank you for these descriptive comments of our work, which we agree are the major key findings. We fully agree that TTF1 analysis suggests that the memory of adenocarcinoma fate even in tumors that have fully squamous morphology.

Arguably, while the effects in the mouse models are clear, the effects of lowering H3K27me3 in the few human cases shown is much weaker, which may limit the extrapolation and importance for human SCC in NSCLC.

Response: We agree that our human data in the original submission was limited. To address this concern, we procured many more patient slides, including 6 pathologist-confirmed cases of adenosquamous disease. Using these samples, we compared EZH2 and H3K27me3 staining intensities in the same tissue sections. Consistently, adenocarcinoma cells had very high H3K27me3 levels and squamous regions had lower H3K27me3 levels, but not absent. In contrast, adenocarcinoma regions had low to absent EZH2 levels, with rare cells expressing high levels, while squamous tumors often had

regions of many cells with high EZH2 levels, though they could still have regions that were negative for the marker, suggesting that EZH2 is dispensable for maintenance of the squamous state. These new data are presented in revised Figure 3. These results mirror our results with the murine samples, which we expanded to include not only the KRAS/*Lkb1* genotype, but also *Lkb1/Pten* squamous, *Lkb1/Pten/p53* adenosquamous tumor slides. We observed a similar pattern of staining, with many regions of squamous exhibiting higher EZH2 and lower H3K27me3 than the adenocarcinoma regions.

This is also of importance with regard to the fact that *Lkb1* deletion in KRAS mutant human NSCLC is rare.

Response: We want to emphasize here that we are not claiming to be modeling purely squamous tumors, but rather tumors with plasticity that could initially be diagnosed as adenocarcinoma, but through time would progress to squamous disease. Clinically, KRAS activation and *Lkb1* deletion are often observed together in early stage adenocarcinoma⁸, and KRAS activation is prevalent in adenosquamous tumors⁹. Our data suggest that *LKB1* mutation is also prevalent in adenosquamous tumors. Together these observations suggest that the genotype of these mice is common for NSCLC patients and particularly common in cases of adenosquamous differentiation. Therefore, we hope you agree that we are modeling a clinically relevant and very poor prognosis subset of tumors^{10,11}.

It also remains unclear why *Lkb1* loss promotes (in a subset of tumors) the loss of Eed protein- here more mechanistic insight is required, and indeed one might ask why they did not validate their observation by using Eed conditional mice instead of *Ezh2* floxed mice.

Response: This is an important question which we have spent much time considering. Our current favorite hypothesis for the connection between LKB1 and EED is that loss of LKB1 may cause cells to alter the amount of homocysteine recycled into the methionine pathway. This lowered methionine availability could directly affect the stability of the PRC2 complex members, as has been shown with the S-adenosyl homocysteine hydrolase inhibitor DZNep¹². EZH2 protein could be independently stabilized through a variety of mechanisms, including O-GlcNAcylation¹³. A hypothesis this complex will take many years to address, and we have now begun the process of Stable Isotope labeling of cultures to learn if homocysteine turnover is indeed changed in the squamous tumors, as is suggested by steady state metabolism of *Lkb1/Pten* tumors².

As for using the *Ezh2* floxed mice in lieu of the *Eed* floxed mice, that decision was driven by desire to know if EZH2 was a specific vulnerability of adeno-squamous tumors; EZH2 is the only protein of the complex for which inhibitors are in clinical trials. We have rewritten the text to reflect this point. Our data clearly indicate that EZH2 is not required for growth of adeno-squamous tumors, suggesting that EZH2 is not a viable therapeutic target for this tumor type. In retrospect, *Eed* floxed mice would have been a better model to recapitulate the natural evolution of this tumor type. However, using *Ezh2* mice instead arguably presented more information – it showed the sufficiency of H3K27me3 gene repression loss to drive squamous transformation, and also demonstrated the dispensability of *Ezh2* expression. In future experiments, *Eed* floxed mice could also be compared, though we expect that the results would largely be the same.

In addition, it remains unclear which of the derepressed Polycomb targets are essential for the observed effects towards SCC differentiation.

Response: Many previous studies suggest that p63 and SOX2 are likely drivers of the squamous fate. Several manuscripts have demonstrated that enforced expression of SOX2, either with *Lkb1* loss or with *Pten* and *Cdkn2a* loss, can drive squamous tumors^{14,15}. Likewise, the critical driving roles of p63, especially the deltaN isoform, have been well studied in squamous tumors¹⁶. In lung squamous tumors, it is now appreciated that SOX2 and deltaNp63 work to bind together at many loci in the genome to promote the squamous transcriptional program¹⁷. Given these previous strong studies, we feel comfortable hypothesizing that SOX2 and deltaNp63 are the major key transcription factors driving the transitioned tumors. This hypothesis is supported by transcriptional changes, immunohistochemistry results and our ChIP-sequencing data. Further experiments may explore the requirement of these two players for the squamous phenotype, but we argue that these experiments are outside the scope of the current study.

Notwithstanding these points, it is of interest that now in the *Lkb1*/*Kras* mutant setting also differential effects are seen for loss of PRC2 function, with some similarities to a recent study documenting a pro-oncogenic effect of PRC2 in *KRas* mutant setting to a tumor suppressive effect in the *KRas*/*P53* mutant setting, also involving strongly the tumor microenvironment.

Response: Thank you for this suggestion. Indeed, we have all read the Serresi et al. paper¹⁸ with great interest. We agree that the immune environment may be a key way through which loss of PRC2 gene repression affects tumor biology as a whole. Like us, the authors demonstrate that Polycomb is not essential for tumor formation in the *KRAS* or *KRAS/p53* mouse models of adenocarcinoma. In the *KRAS* tumors, the authors observed an increase in macrophage infiltration after *Eed* deletion. This is distinct from our result with *KRAS* activation and *Lkb1* deletion, which drives some cells to become *Eed* low and to secrete cytokines that ultimately attract neutrophils, rather than macrophages. They also observe an increase in mucinous differentiation in the *KRAS/p53* model after *Eed* deletion. In contrast, our data suggest that *KRAS/Lkb1* cells can take on some mucinous characteristics while Polycomb is still intact, and can fully transition to squamous tumors when Polycomb is lost. Together these data suggest that Polycomb is critical for defining differentiation state, but that genetics (ie *Lkb1* loss) are also required for *KRAS* driven tumor cells to transition from adenocarcinoma to fully stratified squamous cells.

- 1 Chen, X. *et al.* High expression of trimethylated histone H3 at lysine 27 predicts better prognosis in non-small cell lung cancer. *International Journal of Oncology* **43**, 1467-1480, doi:10.3892/ijo.2013.2062 (2013).
- 2 Xu, C. *et al.* Loss of *Lkb1* and *Pten* Leads to Lung Squamous Cell Carcinoma with Elevated PD-L1 Expression. *Cancer Cell* **25**, 590-604, doi:doi.org/10.1016/j.ccr.2014.03.033 (2014).
- 3 Holm, K. *et al.* Global H3K27 trimethylation and EZH2 abundance in breast tumor subtypes. *Molecular Oncology* **6**, 494-506, doi:10.1016/j.molonc.2012.06.002 (2012).
- 4 Chen, S. *et al.* *H2AK119Ub1* and *H3K27Me3* in molecular staging for survival prediction of patients with pancreatic ductal adenocarcinoma. (2014).
- 5 Xu, K. *et al.* EZH2 Oncogenic Activity in Castration Resistant Prostate Cancer Cells is Polycomb-Independent. *Science (New York, N.Y.)* **338**, 1465-1469, doi:10.1126/science.1227604 (2012).
- 6 Shi, B. *et al.* Integration of Estrogen and Wnt Signaling Circuits by the Polycomb Group Protein EZH2 in Breast Cancer Cells. *Molecular and Cellular Biology* **27**, 5105-5119, doi:10.1128/MCB.00162-07 (2007).
- 7 Lee, Shuet T. *et al.* Context-Specific Regulation of NFKB Target Gene Expression by EZH2 in Breast Cancers. *Molecular Cell* **43**, 798-810, doi:10.1016/j.molcel.2011.08.011.

- 8 Skoulidis, F. *et al.* Co-occurring genomic alterations define major subsets of KRAS - mutant lung adenocarcinoma with distinct biology, immune profiles, and therapeutic vulnerabilities. *Cancer Discovery*, doi:10.1158/2159-8290.cd-14-1236 (2015).
- 9 Shi, X. *et al.* Screening for major driver oncogene alterations in adenosquamous lung carcinoma using PCR coupled with next-generation and Sanger sequencing methods. *Scientific Reports* **6**, 22297, doi:10.1038/srep22297 (2016).
- 10 Filosso, P. L. *et al.* Adenosquamous lung carcinomas: A histologic subtype with poor prognosis. *Lung Cancer* **74**, 25-29, doi:10.1016/j.lungcan.2011.01.030 (2011).
- 11 Guo, Y. *et al.* Clinicopathological characteristics and prognosis of patients with adenosquamous lung carcinoma. *Journal of Huazhong University of Science and Technology [Medical Sciences]* **35**, 350-355, doi:10.1007/s11596-015-1436-z (2015).
- 12 Choudhury, S. R. *et al.* (-)-Epigallocatechin-3-gallate and DZNep reduce polycomb protein level via a proteasome-dependent mechanism in skin cancer cells. *Carcinogenesis* **32**, 1525-1532, doi:10.1093/carcin/bgr171 (2011).
- 13 Chu, C.-S. *et al.* O-GlcNAcylation regulates EZH2 protein stability and function. *Proceedings of the National Academy of Sciences* **111**, 1355-1360, doi:10.1073/pnas.1323226111 (2014).
- 14 Ferone, G. *et al.* SOX2 Is the Determining Oncogenic Switch in Promoting Lung Squamous Cell Carcinoma from Different Cells of Origin. *Cancer Cell* **30**, 519-532, doi:10.1016/j.ccell.2016.09.001.
- 15 Mukhopadhyay, A. *et al.* Sox2 Cooperates with Lkb1 Loss in a Mouse Model of Squamous Cell Lung Cancer. *Cell Reports* **8**, 40-49, doi:10.1016/j.celrep.2014.05.036.
- 16 Ramsey, M. R., He, L., Forster, N., Ory, B. & Ellisen, L. W. Physical Association of HDAC1 and HDAC2 with p63 Mediates Transcriptional Repression and Tumor Maintenance in Squamous Cell Carcinoma. *Cancer Research* **71**, 4373-4379, doi:10.1158/0008-5472.can-11-0046 (2011).
- 17 Watanabe, H. *et al.* SOX2 and p63 colocalize at genetic loci in squamous cell carcinomas. *The Journal of Clinical Investigation* **124**, 1636-1645, doi:10.1172/JCI71545.
- 18 Serresi, M. *et al.* Polycomb Repressive Complex 2 Is a Barrier to KRAS-Driven Inflammation and Epithelial-Mesenchymal Transition in Non-Small-Cell Lung Cancer. *Cancer Cell* **29**, 17-31, doi:10.1016/j.ccell.2015.12.006 (2016).

REVIEWERS' COMMENTS:

Reviewer #1 (Remarks to the Author):

The authors have adequately addressed the issues brought up during the review process and I support publication of the paper.

Reviewer #2 (Remarks to the Author):

The authors have added more data on human (and mouse tumors), as requested, to substantiate the relevance of their findings, which has improved the manuscript and its possible implications for human NSCLC. While they have not necessarily responded to some of my other comments with new data, the textual improvements and clarifications also have helped to improve the manuscript substantially. It would have been of much added value if they were able to confirm their hypothesis on motional metabolism as an underlying mechanism, but it can be understood in lieu of the amount of work involved, that this is left for follow up work. In summary, I can now recommend publication of this improved version

REVIEWERS' FINAL COMMENTS, with our response in Blue:

Reviewer #1 (Remarks to the Author):

The authors have adequately addressed the issues brought up during the review process and I support publication of the paper.

Reviewer #2 (Remarks to the Author):

The authors have added more data on human (and mouse tumors), as requested, to substantiate the relevance of their findings, which has improved the manuscript and its possible implications for human NSCLC. While they have not necessarily responded to some of my other comments with new data, the textual improvements and clarifications also have helped to improve the manuscript substantially. It would have been of much added value if they were able to confirm their hypothesis on motional metabolism as an underlying mechanism, but it can be understood in lieu of the amount of work involved, that this is left for follow up work. In summary, I can now recommend publication of this improved version.

We thank both reviewers for their critical assessment and endorsement of our work.